
# Opposition-based learning techniques in metaheuristics: classification, comparison, and convergence analysis

Rihab Lakbichi[1], Farouq Zitouni[1], Saad Harous[2], Aridj Ferhat[1], Abdelhadi Limane[1], Abdulaziz S. Almazyad[3], Guojiang Xiong[4] and Ali Wagdy Mohamed[5,6,7]

[1] Department of Computer Science and Information Technology, University of Kasdi Merbah, Laboratory of Artificial Intelligence and Information Technology, Ouargla, Algeria
[2] Department of Computer Science, College of Computing and Informatics, Sharjah, United Arab Emirates
[3] King Saud University, Department of Computer Engineering, College of Computer and Information Sciences, Riyadh, Saudi Arabia
[4] College of Electrical Engineering, Guizhou Key Laboratory of Intelligent Technology in Power System, Guiyang, China
[5] Faculty of Graduate Studies for Statistical Research, Operations Research Department, Giza, Egypt
[6] Applied Science Private University, Applied Science Research Center, Amman, Jordan
[7] Saveetha School of Engineering, Saveetha Institute of Medical and Technical Sciences, Department of Biosciences, Chennai, India

Corresponding author
Rihab Lakbichi,
lakbichi.rihab@univ-ouargla.dz

## ABSTRACT

In recent years, opposition-based learning (OBL) has emerged as a powerful enhancement strategy in metaheuristic algorithms (MAs), gaining significant attention for its potential to accelerate convergence and improve solution quality. Existing research lacks a structured analysis of how different OBL variants influence optimization performance when integrated into various MAs. This study categorizes and analyzes nine distinct OBL techniques: basic opposition-based learning, quasi-opposition-based learning, generalized opposition-based learning, current optimum opposition-based learning, quasi-reflection opposition-based learning, centroid opposition-based learning, random opposition-based learning, super opposition-based learning, and stochastic opposition-based learning. To systematically assess the effectiveness of these techniques, five widely used OBL variants—basic opposition-based learning, quasi-opposition-based learning, generalized opposition-based learning, current optimum opposition-based learning, quasi-reflection opposition-based learning—were selected for implementation within five well-established MAs: differential evolution, genetic algorithm, particle swarm optimization, artificial bee colony, and harmony search. These hybridized algorithms were evaluated across different integration phases, including the initialization passes and generation updates phase, and in both phases. To experimentally demonstrate the capability of OBL strategies to enhance MAs that face common issues such as slow convergence, limited exploration, and imbalanced exploration-exploitation, we have used 12 benchmark functions from CEC2022 suite. Key performance metrics—including maximum, minimum, mean, standard deviation, and convergence curves—were rigorously analyzed to quantify the improvements introduced by each OBL-enhanced MA. Additionally, a Friedman test was conducted to statistically validate the performance differences among the variants. The results indicate that quasi-reflection opposition-based learning consistently outperforms

other OBL variants, demonstrating superior convergence speed and solution quality across most benchmark functions.

## INTRODUCTION

Optimization is the process of finding the most effective solution from a set of feasible options to a specific problem (*Diwekar, 2020*). Feasible solutions are those that meet all the constraints of the given problem. Optimization techniques are used to address problems in a variety of fields including science (*Vanfossan, 2022*), engineering (*Zhu et al., 2022*), and industry (*Nascimento, Giudici & Guardani, 2000*). It spans a diverse range of problems, from minimizing production costs (*Cheng, Leung & Li, 2015*) and maximizing operational efficiency (*Rodríguez-García et al., 2020*) to optimizing complex networks and systems (*Cancho & Solé, 2003*). Traditional optimization methods, such as linear programming (*Raidl & Puchinger, 2008*) and gradient-based techniques (*Kearney, Thompson & Boley, 1987*), often fall short when dealing with high-dimensional, nonlinear, and multimodal problems. These challenges have led to the development of more sophisticated and adaptable approaches.

Metaheuristic algorithms (MAs) have gained prominence as powerful optimization techniques capable of navigating complex solution spaces effectively. Unlike traditional methods, metaheuristics do not rely on gradient information and are adept at avoiding local optima, making them suitable for a wide array of optimization problems. Examples of MAs include genetic algorithms (*Shiba, Tsuchiya & Kikuno, 2004*), particle swarm optimization (*Ahmed, Zamli & Lim, 2012*), simulated annealing (*Goffe, Ferrier & Rogers, 1994*), and ant colony optimization (*Okdem & Karaboga, 2009*). These algorithms are inspired by natural and biological processes, such as evolution, swarm intelligence, and annealing in metallurgy, providing a robust and flexible framework for optimization.

Evaluating the performance of MAs is crucial, as their effectiveness hinges on their ability to balance exploration (*i.e.*, global search) and exploitation (*i.e.*, local search) of the solution space. Key performance indicators include convergence speed, robustness to varying initial conditions, and the quality of final solutions. Despite their versatility and success, metaheuristics are not without limitations. Common challenges include premature convergence to sub-optimal solutions (*Evans, 1998*), high computational demands (*Peres & Castelli, 2021*), and sensitivity to parameter settings (*Isiet & Gadala, 2020*). Researchers have proposed various enhancement strategies to mitigate these challenges and enhance the performance of MAs, such as chaotic maps (*Yang, Liu & Zhou, 2014*) and phasor theory (*Liu et al., 2023*). One promising approach is incorporating the OBL concept. It leverages the idea that evaluating both a solution and its opposite can accelerate convergence and improve population diversity in evolutionary algorithms. By considering opposite points in the search space, opposition-based learning (OBL) enhances the exploration capabilities of metaheuristics, potentially leading to more robust and higher-quality solutions.

OBL has emerged as a powerful paradigm in computational intelligence, significantly enhancing the performance of various optimization algorithms. The concept was first introduced by *Tizhoosh (2005)*, who proposed considering opposite solutions simultaneously with current solutions to improve search efficiency. This foundational idea has been successfully integrated into several optimization techniques, reinforcement learning (*Shokri, Tizhoosh & Kamel, 2006*), artificial neural networks (*Ventresca & Tizhoosh, 2007*), and fuzzy systems (*Tizhoosh & Sahba, 2009*), to enhance their operational efficiency. Since January 2005, this concept has been featured in over 400 publications. These works have been disseminated through conferences, journals, and books within the fields of machine learning and soft computing. Specifically, 60% of these publications are journal articles, 38% are conference articles, and 2% are books or theses (*Mahdavi, Rahnamayan & Deb, 2018*). Several survey articles have been published, reviewing more than 200 studies investigating and reviewing the OBL concept (*Al-Qunaieer, Tizhoosh & Rahnamayan, 2010*; *Xu et al., 2014*; *Rojas-Morales, Rojas & Ureta, 2017*; *Mahdavi, Rahnamayan & Deb, 2018*). These surveys provide comprehensive insights into the implementation and effectiveness of OBL across various domains, highlighting its potential to improve optimization algorithms significantly. As the field continues to evolve, ongoing research and development efforts aim to further refine OBL techniques, ensuring their applicability to a broader range of complex optimization problems.

The main contributions of our study can be summarized as follows:

- A systematic classification and analysis of nine state-of-the-art OBL variants—basic opposition-based learning (BOBL), quasi-opposition-based learning (QOBL), generalized opposition-based learning (GOBL), current optimum opposition-based learning (COOBL), quasi-reflection opposition-based learning (QROBL), centroid opposition-based learning (COBL), random opposition-based learning (ROBL), super opposition-based learning (SOBL), and stochastic opposition-based learning ($\beta$-OBL)—was conducted to establish a structured framework for their integration into MAs.

- Five widely used OBL techniques (BOBL, QOBL, GOBL, COOBL, and QROBL) were selected and incorporated into five prominent MAs—differential evolution (DE), genetic algorithm (GA), particle swarm optimization (PSO), artificial bee colony (ABC), and harmony search (HS)—to systematically evaluate their impact on optimization performance.

- Different integration strategies, including opposition-based initialization, opposition-based generation updates, and a hybrid approach combining both, were investigated to determine the most effective application stage.

- Twelve benchmark functions from the CEC 2022 suite were used to comprehensively assess algorithm performance based on key metrics, including maximum, minimum, mean, standard deviation, and convergence curves. Additionally, Friedman's test was applied to statistically compare the performance of different algorithmic variants.

- The effectiveness of QROBL as the most impactful OBL variant was demonstrated, consistently enhancing metaheuristic algorithms by improving convergence speed and

solution quality. Furthermore, significant performance improvements of QOBL and COOBL were observed, highlighting the advantages of integrating OBL strategies into optimization techniques.

The remainder of the article is organized as follows: "Opposition-Based Learning" introduces some basic concepts related to the OBL concept. "Opposition-Based Learning Variants" overviews the nine OBL variants, listed earlier, and categorises the state-of-the-art solutions based on this classification. "Experiments, Numerical Results, and Discussion" defines the way these OBL techniques are hybridized with the five selected MAs and compares and analyses the numerical results. Finally, "Conclusion and Perspectives" concludes the article by summarizing the key findings and suggesting some future research directions.

## OPPOSITION-BASED LEARNING

In computational intelligence, optimization algorithms typically start with a set of initial solutions, then proceed to refine them iteratively toward an optimal outcome. Normally, these initial solutions are generated randomly, uniformly covering the entire search space. This initiation method impacts critical operational factors, including computation time and storage complexity, which are influenced by the proximity of these initial guesses to the optimal solution. The OBL approach simultaneously assesses a solution and its opposite, adopting the nearer of the two as the initial starting point. By selecting the closer of the two guesses, optimization algorithms can significantly accelerate convergence speed and enhance the accuracy of solutions. Moreover, this strategy can be continually applied to refine each solution of the current population throughout the swarming process, thereby consistently enhancing efficiency (*Rahnamayan & Wang, 2008*).

To simplify the understanding of OBL, it is essential to clearly define the concept of opposite numbers in 1-dimensional and $D$-dimensional spaces. Definitions 1 and 2 explain these concepts, respectively (*Tizhoosh, 2005*).

**Definition 1.** *Let $x$ be a real number that belongs to the interval $[a, b]$. Its opposite number denoted as $\check{x}$, is calculated using Eq. (1).*

$$\check{x} = a + b - x. \tag{1}$$

**Definition 2.** *Let $\mathbf{x} = (x_1, x_2, \ldots, x_D)$ be a vector of real numbers that belong to the sub-space $[a_i, b_i]^D$. Its opposite vector, denoted as $\check{\mathbf{x}} = (\check{x}_1, \check{x}_2 \ldots, \check{x}_D)$, is calculated using Eq. (2).*

$$\check{x}_i = a_i + b_i - x_i, \quad \forall i \in \{1, \ldots, D\} \tag{2}$$

where $\check{x}_i$ is the opposition-based counterpart of $x_i$, $a_i$ and $b_i$ represent the lower and upper bounds of $x_i$, respectively, and $D$ is the number of variables.

The opposition-based optimization process can be described as follows: Let $\mathbf{x} = (x_1, x_2, \ldots, x_D)$ be a point in a $D$-dimensional space (*i.e.*, candidate solution), and $f(.)$ be an objective function used to measure the fitness value of candidate solutions. According to Definition 2, $\check{\mathbf{x}} = (\check{x}_1, \check{x}_2 \ldots, \check{x}_D)$ is the opposite point of $\mathbf{x} = (x_1, x_2, \ldots, x_D)$. If $f(\check{\mathbf{x}}) < f(\mathbf{x})$, *i.e.*, $\check{\mathbf{x}}$ has a better fitness value, then point $\mathbf{x}$ is replaced with $\check{\mathbf{x}}$; otherwise, we

continue with **x**. Thus, both the point and its opposite are evaluated simultaneously, and the process continues with the fitter one.

## OPPOSITION-BASED LEARNING VARIANTS

We reviewed more than 100 works incorporating various OBL strategies. We identified nine distinct OBL variants: basic opposition-based learning, quasi-opposition-based learning, generalized opposition-based learning, current optimum opposition-based learning, quasi-reflection opposition-based learning, centroid-opposition-based learning, random-opposition-based learning, super-opposition-based learning, and stochastic opposition-based learning. We catalogued the state-of-the-art published works for each variant in a corresponding table, detailing where the OBL technique was integrated within the original algorithm—whether in the initialization phase (*i.e.*, IP), the swarming phase (*i.e.*, SP), or both. The first and second columns of each table provide the work's reference and publishing year, respectively. The third column lists the original MA to which BOBL was applied. The fourth and fifth columns depict the stages where BOBL was integrated, in the sixth column we highlighted the enhancements that OBL contributed to the original algorithm's performance, and in the final column we highlighted the shortcomings of incorporating OBL into the algorithms. However, certain works have successfully integrated OBL without encountering shortcomings. For the upcoming subsections, we consider the $D$-dimensional points $\mathbf{x} = (x_1, x_2, \ldots, x_D)$ and $\check{\mathbf{x}} = (\check{x}_1, \check{x}_2 \ldots, \check{x}_D)$ as the original point and its opposite, respectively.

### Basic opposition-based learning

Based on Definition 2, this approach aims to provide a better chance of finding an optimal solution by considering both the current solution $\mathbf{x}$ and its opposite $\check{\mathbf{x}}$. The opposite solution $\check{\mathbf{x}}$ is calculated using Eq. (2). Since its introduction, BOBL has been applied to various MAs. Table 1 summarizes the reviewed algorithms that have utilized BOBL to enhance their performance.

### Quasi-opposition-based learning

The QOBL concept was first introduced in 2007 in the work published in *Rahnamayan, Tizhoosh & Salama (2007)* as an extension of OBL. This method generates a random location between the centre and the original points using the function $rand()$. The expression of QOBL is given by Eq. (3). Table 2 presents a summary of algorithms reviewed for their use of QOBL to boost performance.

$$\check{x}_i = \begin{cases} \text{rand}(\frac{a_i+b_i}{2}, a_i + b_i - x_i), & x_i < \frac{a_i+b_i}{2} \\ \text{rand}(a_i + b_i - x_i, \frac{a_i+b_i}{2}), & x_i \geq \frac{a_i+b_i}{2} \end{cases}, \quad \forall i \in \{1, \ldots, D\}. \tag{3}$$

### Generalized opposition-based learning

The GOBL idea combines BOBL and the Cauchy mutation (*i.e.*, random weights) (*Wang et al., 2011*), which may help trapped solutions to jump out of local minima. Equation (4) is employed to compute the opposite solution. The scalar $\delta$ is sampled from the Cauchy

**Table 1  Summary of MAs utilizing BOBL for performance enhancement.**

| Work | Year | Original MA | Place IP | Place SP | Improvements | Shortcomings |
|------|------|-------------|---------|---------|--------------|--------------|
| *Rahnamayan, Tizhoosh & Salama (2006)* | 2006 | Differential evolution | • | | Improved convergence rate | Lack of smart jumping mechanism |
| *Han & He (2007)* | 2007 | Particle swarm optimization | • | | Improved convergence speed; Improved global search ability | Dependence on topology configuration |
| *Malisia & Tizhoosh (2007)* | 2007 | Ant colony system | | • | Improved exploration efficiency | Performed worse than the original algorithm |
| *Rahnamayan, Tizhoosh & Salama (2008)* | 2008 | Differential evolution | • | | Improved convergence rate | Dependence on dimensionality and population size |
| *Dong & Wang (2009)* | 2009 | Differential evolution | • | | Improved convergence rate | Performance not always dominant |
| *Ali, Pant & Abraham (2009)* | 2009 | Hybrid ant colony differential evolution | • | | Improved convergence rate | Limited improvement in solution quality |
| *El-Abd (2011)* | 2011 | Artificial bee colony | • | | Improved solution quality speed | Inconsistent convergence speed |
| *Chatterjee, Ghoshal & Mukherjee (2012)* | 2012 | Harmony search | • | | Improved convergence rate | |
| *Kazemi, Ahmadi & Talebi (2013)* | 2013 | Ant colony optimization | | • | Improved convergence rate | Limited pheromone management strategies |
| *Ajayan & Balaji (2013)* | 2013 | Artificial bee colony | | • | Accelerate conveyance speed | Required parameter adaptations |
| *Saha et al. (2013)* | 2013 | Bat algorithm | | • | Improved convergence speed | Increased computational complexity |
| *Kaucic (2013)* | 2013 | Particle swarm optimization | | • | Improved search efficiency | Experienced higher computational cost |
| *Niu et al. (2014)* | 2014 | Harmony search | • | | Improved diversity; Avoided local optimum | |
| *Banerjee, Mukherjee & Ghoshal (2014)* | 2014 | Harmony search | | • | Improved the convergence rate | Less effective on certain benchmark functions functions |
| *Bhowmik & Chakraborty (2015)* | 2015 | Gravitational search algorithm | | • | Accelerated convergence rate; Improved solutions | |
| *Yang, Xijia & Deng (2015)* | 2015 | Particle swarm optimization | | • | Improved convergence speed | |
| *Yu et al. (2015)* | 2015 | Firefly algorithm | • | | Improved convergence; Local optima avoidance | |
| *Bharti & Singh (2016)* | 2016 | Particle swarm optimization | • | | Improved diversity | Increased computational complexity |
| *Ahandani (2016)* | 2016 | Shuffled bidirectional differential evolution | | • | Improved convergence speed; Premature convergence avoidance | Increased function evaluations |
| *Sarkhel et al. (2017)* | 2017 | Harmony search | | • | Accelerated convergence speed | Converged to a local optimum instead of exploring the global solution |
| *Abd Elaziz, Oliva & Xiong (2017)* | 2017 | Sine cosine algorithm | • | | Improved exploration ability | Time complexity depends on the population size |
| *Ewees, Abd Elaziz & Houssein (2018)* | 2018 | Grasshopper optimization algorithm | • | | Improved convergence rate | Performance is influenced by population size parameter |

| Work | Year | Original MA | Place IP | SP | Improvements | Shortcomings |
|---|---|---|---|---|---|---|
| Gupta & Deep (2019) | 2019 | Sine cosine algorithm | | • | Improved diversity | Increased computational time |
| Shekhawat & Saxena (2020) | 2020 | Crow search algorithm | • | | Improved exploration capability | |
| Gupta et al. (2020) | 2020 | Harris Hawks optimization | | • | Improved exploration capability | |
| Zhao et al. (2020) | 2020 | Salp swarm algorithm | • | | Improved convergence speed / Improved exploration ability | Performance is problem-dependent |
| Xu et al. (2022) | 2022 | Fish migration optimization | | • | Improved exploration capability | |
| Si, Miranda & Bhattacharya (2022) | 2022 | Salp swarm algorithm | • | | Improved convergence speed | |
| Sahoo et al. (2023) | 2023 | Bee colony optimization | • | | Improved exploration capability / Improved convergence rate | Underperformed in some problem applications |
| Joshi (2023) | 2023 | Gravitational search algorithm | • | | Accelerated convergence / Improved search capability | |
| Abualigah, Diabat & Elaziz (2023) | 2023 | Slime Mould algorithm | • | | Accelerated convergence speed / Premature convergence avoidance | |
| Chen et al. (2024) | 2024 | Ant colony path planning optimization | • | | Increased convergence rate | |
| Zhao et al. (2024) | 2024 | Brain storm optimization | | • | Increased search density / Local optimum avoidance | OBL has no significant impact on the algorithm |
| Wang, Li & Yan (2024) | 2024 | Differential evolution | • | | Adaptive adjusting of sub-population size | |
| Tian et al. (2024) | 2024 | Particle swarm optimization | • | | Accelerated convergence process | |
| Pham, Nguyen Dang & Nguyen (2024) | 2024 | Sine cosine algorithm | • | | Improved balance between exploration and exploitation | |

**Table 2   Summary of MAs utilizing QOBL for performance enhancement.**

| Work | Year | Original MA | Place | | Improvements | Shortcomings |
|---|---|---|---|---|---|---|
| | | | IP | SP | | |
| Rahnamayan, Tizhoosh & Salana (2007) | 2008 | Differential evolution | • | | Improved convergence speed | |
| Mandal & Roy (2013) | 2013 | Teaching learning based optimization | • | • | Accelerated convergence speed | |
| Sultana & Roy (2014) | 2014 | Teaching learning based optimization | • | | Accelerated convergence speed | |
| Bhattacharjee, Bhattacharya & Nee Dey (2014) | 2014 | Real coded chemical reaction optimization | • | | Improved convergence rate | Required careful parameter tuning |
| Yazdani & Shanbehzadeh (2015) | 2015 | Cartesian genetic programming | • | | Enhanced exploration ability | Dependence on Parameter setting |
| Roy & Paul (2015) | 2015 | Gravitational search algorithm | • | • | Improved solution quality computational speed | Failed to outperform all competitors |
| Shiva & Mukherjee (2015a, 2015b) | 2015 | Harmony Search | • | | Accelerated convergence speed | Improved accuracy depends on specific cases |
| Basu (2016a) | 2016 | Differential evolution | • | | Improved convergence speed | Required high computational time |
| Basu (2016b) | 2016 | Group search optimization | • | | Improved effectiveness Enhanced solutions | Required high computational time |
| Shankar & Mukherjee (2016) | 2016 | Harmony search | • | | Improved performance | Accuracy of some cases not significantly improved |
| Guha, Roy & Banerjee (2017) | 2017 | Symbiotic organism search algorithm | • | • | Local optima avoidance Accelerated convergence speed | |
| Hazra & Roy (2019) | 2019 | Chemical reaction optimization | • | | Accelerated convergence speed | No significant enhancement compared to the original algorithm |
| Truong et al. (2019) | 2019 | Symbiotic organisms search | • | | Improved solutions Improved convergence speed | |
| Chen, Li & Yang (2020) | 2020 | Whale optimization algorithm | • | | Local optima avoidance | No significant improvement in some text functions Increased execution time comparatively |
| Elsisi (2022) | 2022 | Grey wolf optimizer | • | | Enhanced exploration and exploitation Local optimum avoidance | |
| Wang et al. (2022) | 2022 | Grey wolf optimizer | • | | Balanced exploration and exploitation | Required high computational complexity |
| Chaudhuri & Sahu (2022) | 2022 | Jaya algorithm | • | | Improved exploration capability | Failed to consistently deliver superior results across all datasets |
| Çelik (2023) | 2023 | Arithmetic optimization | • | | Enhanced convergence rate Local minima avoidance | |
| Zhao et al. (2023) | 2023 | Marine predators algorithm | • | | Increased diversity Local minima avoidance | Required additional strategies for further improvements |
| Eirgash & Toğan (2023) | 2023 | Teaching learning strategy | • | • | Enriched diversity Local optimal avoidance | |
| Sahoo et al. (2024) | 2024 | Moth flame optimization | • | | Enhanced performance | |
| Chandran & Mohapatra (2024) | 2024 | Tunicate swarm algorithm | • | • | Accelerated convergence Improved exploration ability | Getting stuck in local optima in high-dimensional problems |

distribution. In Table 3, we summarize the algorithms reviewed that employed GOBL for performance enhancement.

$$\check{x}_i = \delta \times (a_i + b_i) - x_i, \quad \forall i \in \{1, \ldots, D\}. \tag{4}$$

### Current optimum opposition-based learning

Another enhancement of OBL is called COOBL (*Yang, 2017*). It uses the search information of the current best solution. The mathematical formula of COOBL is given by Eq. (5). The $D$-dimensional point $\mathbf{x}^*$ denotes the best solution within the current population. Table 4 provides an overview of the reviewed algorithms that incorporated COOBL to enhance performance.

$$\check{x}_i = 2 \times x_i^* - x_i, \quad \forall i \in \{1, \ldots, D\}. \tag{5}$$

### Quasi-reflection opposition-based learning

The QROBL technique is an extension of QOBL (*Ergezer, Simon & Du, 2009*). It generates a random $D$-dimensional point between the centre and $\mathbf{x}$ points. Its algebraic expression is formulated by Eq. (6). The algorithms that utilized QROBL for performance improvement are summarized in Table 5.

$$\check{x}_i = \begin{cases} \text{rand}(x_i, \frac{a_i+b_i}{2}), & x_i < \frac{a_i+b_i}{2} \\ \text{rand}(\frac{a_i+b_i}{2}, x_i), & x_i \geq \frac{a_i+b_i}{2} \end{cases}, \quad \forall i \in \{1, \ldots, D\}. \tag{6}$$

### Centroid opposition-based learning

The COBL variant replaces the current optimum in COOBL by the location of the centroid (*Rahnamayan et al., 2014*). It is mathematically expressed by Eq. (7). The $D$-dimensional point $\mathbf{m}$ denotes the centre of gravity of the current population. It is calculated using the expression $\sum_{k=1}^{N} \mathbf{x}_k$, where $N$ is the population size. Table 6 outlines the reviewed algorithms that have applied COBL to improve performance.

$$\check{x}_i = 2 \times m_i - x_i, \quad \forall i \in \{1, \ldots, D\}. \tag{7}$$

### Random opposition-based learning

The ROBL variant is another enhancement of BOBL, where the original point is multiplied by random numbers to improve exploration (*Long et al., 2019*; *Bairathi & Gopalani, 2020*). It is presented in Eq. (8). Table 7 summarises the reviewed algorithms that used ROBL for performance enhancement.

$$\check{x}_i = a_i + b_i - \text{rand}(0, 1) \times x_i, \quad \forall i \in \{1, \ldots, D\}. \tag{8}$$

### Super opposition-based learning

In SOBL (*Tizhoosh & Ventresca, 2008*), the opposite point $\check{\mathbf{x}}$ can be defined based on the distance of $\mathbf{x}$ from the centre of the interval. It is expressed by Eq. (9). In Table 8, we provide a summary of algorithms that have used SOBL to enhance their performance.

**Table 3 Summary of MAs utilizing GOBL for performance enhancement.**

| Work | Year | Original MA | Place | | Improvements | Shortcomings |
|------|------|-------------|-------|-----|--------------|--------------|
| | | | IP | SP | | |
| *Wang, Wu & Rahnamayan (2011)* | 2011 | Differential evolution | • | • | Improved convergence rate | Underperformed on shifted and large-scale problems |
| *Wang et al. (2011)* | 2011 | Particle swarm optimization | • | • | Improved convergence speed | |
| *Wang, Rahnamayan & Wu (2013)* | 2013 | Differential evolution | • | • | Improved quality of solutions | |
| *Si, De & Bhattacharjee (2014)* | 2014 | Particle swarm optimization | | • | Enhanced solution quality | Exhibits a slower convergence rate compared to other competitive algorithm |
| *Wang (2015)* | 2015 | Artificial bee colony | • | • | Balanced exploration and exploitation | Failed to perform optimally across all problem functions |
| *Wei et al. (2016)* | 2016 | Differential evolution | • | • | Improved convergence speed | |
| *Chen et al. (2016)* | 2016 | Teaching learning based optimization | • | • | Enhanced convergence speed | Ranking second in terms of objective value |
| *Guo et al. (2017)* | 2017 | Harmony search | | • | Enhanced exploitation capability | |
| *Wang et al. (2017)* | 2017 | Suckoo search | | • | Accelerated convergence speed | |
| *Deng et al. (2022)* | 2022 | Differential evolution | • | • | Improved convergence speed Increased diversity | Required high computational time for high-dimensional complex problems |
| *Chen et al. (2023)* | 2023 | Equilibrium optimizer | | • | Local optima avoidance | Required increased computational time |
| *Qiu et al. (2024)* | 2024 | Harmony search | | • | Enhanced convergence rate | |
| *Zhang et al. (2024)* | 2024 | Particle swarm optimization | • | • | Enhanced convergence rate | Required slightly high computation time |

**Table 4 Summary of MAs utilizing COOBL for performance enhancement.**

| Work | Year | Original MA | Place | | Improvements | Shortcomings |
|------|------|-------------|-------|-----|--------------|--------------|
| | | | IP | SP | | |
| *Xu et al. (2011)* | 2011 | Differential evolution | • | • | Improved performance | Failed to achieve best results across all types of test functions |
| *Cao, Li & Wang (2013)* | 2013 | Animal migration optimization | • | • | Improved searchability Accelerated convergence rate | No significant improvement for certain type of optimization problems |
| *Chen & Tang (2015)* | 2015 | Particle swarm optimization | • | | Improved convergence rate | Relied on parameter settings for optimal performance |
| *Mahdavi, Rahnamayan & Deb (2016)* | 2016 | Differential evolution | • | • | Enhanced solutions | |
| *Yang (2017)* | 2017 | Differential evolution | • | • | Improved convergence rate | |
| *Azmi et al. (2019)* | 2019 | Simulated Kalman filter | | • | Increased exploration | |

$$\check{x}_i \in \begin{cases} [a_i, x_i), & x_i > \frac{a_i+b_i}{2} \\ [a_i, x_i) \cup (x_i, b_i], & x_i = \frac{a_i+b_i}{2}, \quad \forall i \in \{1, \ldots, D\}. \\ (a_i, b_i], & x_i < \frac{a_i+b_i}{2}, \end{cases} \tag{9}$$

**Table 5 Summary of MAs utilizing QROBL for performance enhancement.**

| Work | Year | Original MA | Place | | Improvements | Shortcomings |
|------|------|-------------|-------|-----|--------------|--------------|
| | | | IP | SP | | |
| *Ergezer, Simon & Du (2009)* | 2009 | Biogeography-based optimization | • | • | Improved performance | |
| *Bhattacharya & Chattopadhyay (2010)* | 2010 | Biogeography-based optimization | • | • | Accelerated convergence rate<br>Improved solutions | |
| *Ergezer & Simon (2015)* | 2015 | Evolutionary algorithms | • | • | Accelerated performance | |
| *Das, Bhattacharya & Chakraborty (2018)* | 2018 | Ions motion optimization | • | • | Improved exploration and exploitation | |
| *Luo & Yu (2022)* | 2022 | Cuckoo search | • | | Improved convergence speed | |
| *Nama (2022)* | 2022 | Slime mold algorithm | • | • | Improved convergence rate<br>Local optima avoidance | Required additional strategies for further improvement |
| *Dhal et al. (2023)* | 2023 | Aquila optimizer | • | • | Improved exploitation | Failed to perform effectively in real-world applications |
| *Bacanin et al. (2023)* | 2023 | Arithmetic optimization algorithm | • | • | Improved diversity | Required high computational cost |
| *Chauhan et al. (2024)* | 2024 | Arithmetic optimisation algorithm | • | | Local optima avoidance | The convergence efficiency is limited in some test functions |
| *Sahoo et al. (2024)* | 2024 | Moth flame optimization | • | • | Improved performance | Failed to achieve superior optimal results for many objectives |

**Table 6 Summary of MAs utilizing COBL for performance enhancement.**

| Work | Year | Original MA | Place | | Improvements | Shortcomings |
|------|------|-------------|-------|-----|--------------|--------------|
| | | | IP | SP | | |
| *Rahnamayan et al. (2014)* | 2014 | Differential evolution | • | • | Enhanced learning<br>Improved convergence rate | Failed to outperform in all test functions |
| *Zhou, Ding & Lei (2018)* | 2018 | Firefly algorithm | • | • | Improved exploration ability | Required careful parameter tuning for optimal results |
| *Liao & Zhou (2019)* | 2019 | Grasshopper optimization algorithm | | • | Improved exploration and exploitation | |
| *Zhou et al. (2019)* | 2019 | Firefly algorithm | • | • | Improved convergence rate | |
| *Si & Bhattacharya (2021)* | 2021 | Sine cosine algorithm | • | • | Improved exploration | |
| *Xiang & Wu (2023)* | 2023 | Hybrid Salp swarm and butterfly optimization algorithm | • | • | Accelerated convergence speed | |

## Stochastic opposition-based learning

Most OBL variants compute opposite solutions based on the uniform distribution. However, $\beta$-OBL utilizes the Beta distribution, *i.e.*, $\beta(\alpha, \beta)$, to calculate concave or convex opposite solutions (*Choi, Togelius & Cheong, 2021*). We compute the opposite point, $\check{x}$, using Eq. (10). The variables spread and mode are computed differently for the former and the latter case, respectively. The symbol $N$ denotes the population size. The symbol $x_j^{(i)}$ represents the $i$-th solution. The term $N(0, 0.5)$ samples a random number from the

**Table 7 Summary of MAs utilizing ROBL for performance enhancement.**

| Work | Year | Original MA | Place | | Improvements | Shortcomings |
|------|------|-------------|-------|------|--------------|--------------|
| | | | IP | SP | | |
| *Long et al. (2019)* | 2019 | Grey wolf optimizer | | • | Enhanced diversity<br>Local optima avoidance | Required high number of function evaluations |
| *Zhang & Zhang (2021)* | 2021 | Sparrow search algorithm | • | • | Enhanced exploration ability | |
| *Ali, Fathimathul Rajeena & Salama Abd Elminaam (2022)* | 2022 | Artificial hummingbird algorithm | | • | Enhanced exploitation | Outperformed by some methods in terms of speed |
| *Ma et al. (2022)* | 2022 | Moth-flame optimization | • | • | Increased diversity<br>Improved search speed | Required high computational time |
| *Balakrishnan, Dhanalakshmi & Mahadeo Khaire (2022)* | 2022 | Marine predators algorithm | • | | Enhanced exploration | |

**Table 8 Summary of MAs utilizing SOBL for performance enhancement.**

| Work | Year | Original MA | Place | | Improvements | Shortcomings |
|------|------|-------------|-------|------|--------------|--------------|
| | | | IP | SP | | |
| *Kaucic (2013)* | 2013 | Particle swarm optimization | | • | Avoided premature convergence | |
| *Razak, Nasir & Abd Ghani (2022)* | 2022 | Spiral dynamic algorithm | • | • | Avoided premature convergence | Increased computational cost |
| *Abdul Razak et al. (2024)* | 2024 | Manta Ray foraging algorithm | • | • | Enhanced exploration and exploitation | |

Gaussian distribution. Table 9 summarizes the reviewed algorithms that implemented $\beta$-OBL to enhance their performance.

$$\check{x}_i = \beta(\alpha, \beta) \times (a_j - b_j) + b_j, \quad \forall i \in \{1, \dots, D\} \tag{10}$$

$$\alpha = \begin{cases} \text{spread} \times \text{peak}, & \text{mode} < 0.5 \\ \text{spread}, & \text{otherwise} \end{cases}, \quad \beta = \begin{cases} \text{spread}, & \text{mode} < 0.5 \\ \text{spread} \times \text{peak}, & \text{otherwise} \end{cases} \tag{11}$$

$$\text{peak} = \begin{cases} \frac{(\text{spread}-2) \times \text{mode}+1}{\text{spread} \times (1-\text{mode})}, & \text{mode} < 0.5 \\ \frac{2-\text{spread}}{\text{spread}} + \frac{\text{spread}-1}{\text{spread} \times \text{mode}}, & \text{otherwise} \end{cases} \tag{12}$$

$$\text{normDiv} = \frac{1}{N} \sum_{i=1}^{N} \sum_{j=1}^{D} \sqrt{\frac{1}{D} \left( \frac{x_j^{(i)} - \bar{x}_j}{a_j - b_j} \right)^2}, \quad \bar{\mathbf{x}} = \frac{1}{N} \sum_{i=1}^{N} \mathbf{x}_i \tag{13}$$

$$\text{spread} = \begin{cases} \left( \frac{1}{\sqrt{\text{normDiv}}} \right)^{1+N(0,0.5)}, & \text{concave} \\ 0.1 \times \sqrt{\text{normDiv}} + 0.9 & \text{convex} \end{cases}, \quad \text{mode} = \begin{cases} \frac{a_j - x_{i,j}}{a_j - b_j}, & \text{concave} \\ \text{mode} = \frac{x_{i,j} - b_j}{a_j - b_j}, & \text{convex} \end{cases}. \tag{14}$$

After conducting an extensive literature review on the applications and effectiveness of OBL techniques in metaheuristic optimization, we have identified and summarized four fundamental limitations. Firstly, many OBL techniques exhibit a high degree of similarity, differing mainly in how the opposition value is selected, which results in their effectiveness

**Table 9 Summary of MAs utilizing $\beta$-OBL for performance enhancement.**

| Work | Year | Original MA | Place | | Improvements | Shortcomings |
|------|------|-------------|-------|----|--------------|--------------|
| | | | IP | SP | | |
| *Park & Lee (2015)* | 2015 | Differential evolution | • | • | Enhanced convergence speed | Performance is highly sensitive to jumping rate tuning |
| | | | | | Enhanced searchability | |
| *Choi, Togelius & Cheong (2021)* | 2021 | Differential evolution | • | • | Reduced computational cost | |
| *Choi & Pachauri (2024)* | 2024 | Differential evolution | • | • | Enhanced performance | |
| *Zitouni et al. (2024)* | 2024 | Great wall construction algorithm | • | • | Enhanced exploration and exploitation | Computational cost slightly increased |
| | | | | | Local optima avoidance | |

being highly problem-dependent. This means that while an OBL variant may show significant improvement for a particular optimization problem with a specific search space contour, it may not generalize well to other problems with different landscape characteristics. Secondly, despite various OBL techniques demonstrating promising results within specific metaheuristic frameworks, their scalability to other MAs remains largely unproven. Most studies focus on enhancing a single algorithm, such as DE or PSO, without confirming whether the same OBL technique would be equally effective in another optimization algorithm. Thirdly, traditional OBL approaches generate only a single opposite solution for each candidate solution, which may not be sufficient for effectively exploring the search space. By considering only one opposite candidate, OBL methods might limit diversity and fail to fully exploit the best-known solutions, especially in high-dimensional or complex landscapes where having only two values—original and opposite—may not provide sufficient diversity to explore better regions. Finally, in cases involving symmetric problems, the use of OBL may be counterproductive, as the generated opposite solution could have the same fitness value as the original candidate. This redundancy results in wasted computational resources and does not contribute to accelerating convergence or improving solution quality. In some scenarios, it may even mislead the search process, causing the global optimum to be missed. These four limitations highlight the need for more adaptive and generalized OBL strategies that can effectively address problem dependency, ensure cross-algorithm scalability, enhance search diversity, and mitigate inefficiencies in symmetric problem landscapes.

## EXPERIMENTS, NUMERICAL RESULTS, AND DISCUSSION

In this section, we delve into the performance analysis of our comparative study, focusing on integrating five widely used OBL techniques into the operational framework of five well-known MAs. It aims to determine the most effective OBL variant and which combination yields the best outcome. The methodology for integrating the OBL techniques into the five selected MAs is detailed in "Applications of OBL in MAs". "Benchmark Functions and Parameter Settings" provides an overview of the benchmark

functions utilized in our comparative study and details the parameter settings for each MA. "Experimental Results and Discussion" details the comparative research and thoroughly discusses the numerical results using three main metrics: optimality of solutions, execution time, and number of function evaluations.

## Applications of OBL in MAs

In our experiments, we integrated five OBL variants (BOBL (*Tizhoosh, 2005*), QOBL (*Rahnamayan, Tizhoosh & Salama, 2007*), GOBL (*Wang et al., 2011*), COOBL (*Xu et al., 2011*), and QROBL (*Ergezer, Simon & Du, 2009*)) with five well-established MAs (DE (*Storn & Price, 1997*), PSO (*Kennedy & Eberhart, 1995*), GA (*Babović & Wu, 1975*), ABC (*Karaboga, 2005*), and HS (*Yang, 2009*)). Each OBL technique was incorporated into these algorithms at various stages, including the IP, the SP, or both, resulting in a total of 75 different algorithms. The selected OBL variants were chosen based on their demonstrated superior performance in numerous studies, proving to be more effective in improving optimization algorithms compared to other existing methods. The chosen MAs were selected for their proven success in the literature, as well as their flexibility and adaptability, which facilitate the seamless integration of OBL strategies without significant alterations to the algorithms' structures. Several studies have highlighted that the combination of OBL techniques with these algorithms leads to notable performance enhancements, as evidenced by their success in various benchmark problems. To the best of our knowledge, this study is the first to comprehensively compare these combinations, offering a detailed analysis of their comparative performance.

To streamline the memorization and easy identification of the distinct features across various combinations, each combination will be represented using a simple notation: X-Y-Z. In this notation, X denotes the applied OBL variant, which can be BOBL, QOBL, GOBL, COOBL, or QROBL. Y signifies the chosen MA, which can also be DE, PSO, GA, ABC, or HS. Lastly, Z represents the selected phase, indicated by the acronyms IP, SP, or ISP. For example, the representation BOBL-PSO-IP indicates that BOBL was applied during the initialization phase of the first population in the PSO algorithm.

## Benchmark functions and parameter settings

To evaluate the performance of the different combinations, a set of 12 challenging benchmark functions derived from the CEC 2022 benchmark suite (*Yy, 2024*) has been employed. These benchmark functions serve as a rigorous test bed for verifying the performance of each combination. The specifics of these functions are detailed in Table 10. The dimension of the search space is set to 10 and 20.

Table 11 summarizes the parameter settings for the various MAs chosen to evaluate the performance of the selected OBL techniques. These parameters were meticulously chosen based on empirical testing and recommendations from existing literature to ensure optimal performance for each algorithm. In our experiments, the population size and the maximum number of generations are set to 50 and 1,000, respectively.

**Table 10 Characteristics of the 12 test functions.**

| Function | Function name | Optimal solution |
|---|---|---|
| **Unimodal functions** | | |
| $F_1$ | Shifted and full Rotated Zakharov function | 300 |
| **Basic/Multimodal functions** | | |
| $F_2$ | Rotated Rosenbrock's function | 400 |
| $F_3$ | Rotated expanded Schaffer's $f_6$ function | 600 |
| $F_4$ | Rotated non-continuous Rastrigin's function | 800 |
| $F_5$ | Rotated Levy function | 900 |
| **Hybrid functions** | | |
| $F_6$ | Hybrid function 1 ($N = 3$) | 1,800 |
| $F_7$ | Hybrid function 2 ($N = 6$) | 2,000 |
| $F_8$ | Hybrid function 3 ($N = 5$) | 2,200 |
| **Composition functions** | | |
| $F_9$ | Composition function 1 ($N = 5$) | 2,300 |
| $F_{10}$ | Composition function 2 ($N = 4$) | 2,400 |
| $F_{11}$ | Composition function 3 ($N = 5$) | 2,600 |
| $F_{12}$ | Composition function 4 ($N = 6$) | 2,700 |
| **Search range:** $[-100, 100]^D$ | | |

## Experimental results and discussion

We employed a range of evaluation metrics to assess the performance of the various MAs and OBL techniques used in our comparative study, ensuring a comprehensive analysis. The selected metrics are given as follows:

1. Average fitness value (mean): This metric represents the mean fitness value obtained from each MA averaged over the specified number of runs. We used this metric to understand how well the algorithm performs on average.

2. Minimum fitness value (best): This metric indicates the lowest fitness value calculated from each MA across the designated number of runs. It gives insight into the algorithm's capability to reach optimal or near-optimal solutions.

3. Maximum fitness value (worst): This metric shows the highest fitness value computed from each MA across the designated number of runs. It helps in evaluating the consistency and reliability of the algorithm.

4. Standard deviation value (STD): This metric assesses the performance of MAs because it measures the variability and consistency of the algorithms' results across multiple runs, indicating their reliability and robustness. It is worth pointing out that the mathematical expression used to compute its value is given by Eq. (15).

$$\text{STD} = \frac{1}{|R|} \sum_{i=1}^{|R|} \left( f(\mathbf{s}_i) - f(\mathbf{s}^*) \right)^2 \tag{15}$$

where $|R|$ is the number of runs, $\mathbf{s}_i$ the best-obtained solution at iteration $i$, and $\mathbf{s}^*$ is the optimal solution.

**Table 11 Parameter settings of the MAs used for the comparative study.**

| Algorithm | Parameter | Value |
|---|---|---|
| DE | Crossover rate | 0.2 |
| | Scaling factor | (0.3, 0.9) |
| GA | Crossover percentage | 0.7 |
| | Mutation percentage | 0.3 |
| | Mutation rate | 0.1 |
| PSO | Inertia weight | 1 |
| | Inertia weight damping ratio | 0.99 |
| | Personal learning coefficient | 1.5 |
| | Global learning coefficient | 2.0 |
| ABC | Trial limit | $N \times D$ |
| | Food number | $0.5 \times N$ |
| HS | Harmony memory consideration rate | 0.9 |
| | Pitch adjustment rate | 0.45 |
| | Bandwidth | 0.02 |

5. Execution time (ET): This metric refers to the time a given MA takes to complete its optimization process. This metric is important for understanding which algorithm quickly converges to the optimal solution.

6. Number of function evaluations (NFEs): This metric refers to the total number of times the objective function is called by the considered MA to evaluate the candidate solutions. We apply it to compare the efficiency of algorithms, as it directly reflects the computational effort required to achieve a solution.

7. Convergence rate (CR): This measure evaluates how quickly the solutions improve over iterations. It helps understand how quickly an algorithm converges to the optimal solution.

Finally, due to the stochastic nature of MAs, each algorithm was run 30 times—to obtain reliable and consistent results—using MATLAB R2023a, a widely recognized platform for numerical computation and algorithm development. The computational experiments were conducted on a system equipped with a 12th Gen Intel(R) Core(TM) i5-1235U processor running at 1.30 GHz, paired with 16 GB of RAM. This hardware configuration runs on Windows 11.

### Performance analysis

In this section, we analyze and discuss the performance of different combinations, highlighting which one yields the best results. Tables 12, 13, 14, 15, and 16 provide a detailed comparison of the performance of the considered five MAs along with their OBL variants across 12 benchmark functions, all evaluated at a dimension of 10. In parallel, Tables 17, 18, 19, 20, and 21 display the performance results for the same set of algorithms and their OBL variants, but at an increased dimension of 20. This comprehensive

**Table 12  Performance comparison among DE and its variants across 12 functions ($D = 10$).**

| Variants | | STD | ET | NFEs |
|---|---|---|---|---|
| BOBL-DE | IP | 5/3/4 | 12/0/0 | 0/0/12 |
| | SP | 1/3/8 | 0/0/12 | 0/0/12 |
| | IP-SP | 3/3/6 | 0/0/12 | 0/0/12 |
| COOBL-DE | IP | 1/8/3 | 8/0/4 | 0/0/12 |
| | SP | 4/0/8 | 0/0/12 | 0/0/12 |
| | IP-SP | 2/0/10 | 0/0/12 | 0/0/12 |
| GOBL-DE | IP | 6/3/3 | 0/0/12 | 0/0/12 |
| | SP | 6/3/3 | 0/0/12 | 0/0/12 |
| | IP-SP | 6/3/3 | 0/0/12 | 0/0/12 |
| QOBL-DE | IP | 6/3/3 | 6/0/6 | 0/0/12 |
| | SP | 4/3/5 | 0/0/12 | 0/0/12 |
| | IP-SP | 7/3/2 | 0/0/12 | 0/0/12 |
| QROBL-DE | IP | 6/3/3 | 2/2/8 | 0/0/12 |
| | SP | 6/4/2 | 0/0/12 | 0/0/12 |
| | IP-SP | 7/2/3 | 0/0/12 | 0/0/12 |

**Table 13  Performance comparison among GA and its variants across 12 functions ($D = 10$).**

| Variants | | STD | ET | NFEs |
|---|---|---|---|---|
| BOBL-GA | IP | 8/1/3 | 5/2/5 | 0/0/12 |
| | SP | 7/1/4 | 2/1/9 | 0/0/12 |
| | IP-SP | 7/0/5 | 2/1/9 | 0/0/12 |
| COOBL-GA | IP | 2/1/9 | 2/0/10 | 0/0/12 |
| | SP | 2/2/8 | 11/0/1 | 0/0/12 |
| | IP-SP | 3/0/9 | 11/0/1 | 0/0/12 |
| GOBL-GA | IP | 3/0/9 | 5/0/7 | 0/0/12 |
| | SP | 7/1/4 | 2/1/9 | 0/0/12 |
| | IP-SP | 5/0/7 | 2/0/10 | 0/0/12 |
| QOBL-GA | IP | 7/0/5 | 3/0/9 | 0/0/12 |
| | SP | 8/0/4 | 2/0/10 | 0/0/12 |
| | IP-SP | 9/1/2 | 2/ 0/10 | 0/0/12 |
| QROBL-GA | IP | 7/0/5 | 8/1/3 | 0/0/12 |
| | SP | 8/1/3 | 2/0/10 | 0/0/12 |
| | IP-SP | 5/2/5 | 2/0/10 | 0/0/12 |

comparison enables a deeper understanding of how the algorithms and their variants perform across different problem dimensions. These tables focus on three primary metrics: STD, ET, and NFEs, where each one is evaluated across three phases: IP, SP, and IP-SP. The data in the tables are presented in the format A/B/C, where A represents the number of functions in which the variant outperforms the original algorithm, B shows the number of functions where both perform similarly, and C indicates the number of functions where

**Table 14 Performance comparison among PSO and its variants across 12 functions ($D = 10$).**

| Variants | | STD | ET | NFEs |
|---|---|---|---|---|
| BOBL-PSO | IP | 4/1/7 | 0/0/12 | 0/0/12 |
| | SP | 4/3/5 | 0/0/12 | 0/0/12 |
| | IP-SP | 5/2/5 | 0/0/12 | 0/0/12 |
| COOBL-PSO | IP | 3/2/7 | 0/0/12 | 0/0/12 |
| | SP | 0/1/11 | 0/0/12 | 0/0/12 |
| | IP-SP | 0/1/11 | 0/0/12 | 0/0/12 |
| GOBL-PSO | IP | 5/2/5 | 11/0/1 | 0/0/12 |
| | SP | 4/2/6 | 12/0/0 | 0/0/12 |
| | IP-SP | 6/2/4 | 11/0/1 | 0/0/12 |
| QOBL-PSO | IP | 5/1/6 | 12/0/0 | 0/0/12 |
| | SP | 4/1/7 | 1/0/11 | 0/0/12 |
| | IP-SP | 5/2/5 | 0/0/12 | 0/0/12 |
| QROBL-PSO | IP | 4/1/7 | 10/0/2 | 0/0/12 |
| | SP | 2/1/9 | 0/0/12 | 0/0/12 |
| | IP-SP | 4/2/6 | 0/0/12 | 0/0/12 |

**Table 15 Performance comparison among HS and its variants across 12 functions ($D = 10$).**

| Variants | | STD | ET | NFEs |
|---|---|---|---|---|
| BOBL-HS | IP | 4/2/6 | 0/0/12 | 0/0/12 |
| | SP | 5/1/6 | 0/0/12 | 0/0/12 |
| | IP-SP | 6/0/6 | 0/0/12 | 0/0/12 |
| COOBL-HS | IP | 2/0/10 | 0/0/12 | 0/0/12 |
| | SP | 7/0/5 | 0/0/12 | 0/0/12 |
| | IP-SP | 7/0/5 | 0/0/12 | 0/0/12 |
| GOBL-HS | IP | 4/1/7 | 0/0/12 | 0/0/12 |
| | SP | 4/1/7 | 0/0/12 | 0/0/12 |
| | IP-SP | 5/1/6 | 0/0/12 | 0/0/12 |
| QOBL-HS | IP | 6/1/5 | 0/0/12 | 0/0/12 |
| | SP | 5/0/7 | 0/0/12 | 0/0/12 |
| | IP-SP | 3/0/9 | 0/0/12 | 0/0/12 |
| QROBL-DE | IP | 3/0/9 | 3/0/9 | 0/0/12 |
| | SP | 5/1/6 | 0/0/12 | 0/0/12 |
| | IP-SP | 4/0/8 | 0/0/12 | 0/0/12 |

the original algorithm outperforms the variant. These tables offer a more concise version of the detailed data found in the Appendix, providing a clearer summary of the performance results.

Table 12 presents a detailed comparison of several DE variants across three performance metrics—STD, ET, and NFEs—for 12 functions with $D = 10$. In the following, we summarize the key insights and analysis of these results:

**Table 16 Performance comparison among ABC and its variants across 12 functions ($D = 10$).**

| Variants | | STD | ET | NFEs |
|---|---|---|---|---|
| BOBL-ABC | IP | 7/0/5 | 7/0/5 | 1/11/0 |
| | SP | 7/0/5 | 10/0/2 | 0/0/12 |
| | IP-SP | 6/0/6 | 9/0/3 | 0/0/12 |
| COOBL-ABC | IP | 1/0/11 | 12/0/0 | 0/12/0 |
| | SP | 5/1/6 | 10/0/2 | 0/0/12 |
| | IP-SP | 6/0/6 | 8/0/4 | 0/0/12 |
| GOBL-ABC | IP | 6/1/5 | 12/0/0 | 0/12/0 |
| | SP | 7/0/5 | 9/0/3 | 0/0/12 |
| | IP-SP | 7/0/5 | 9/ 0/3 | 0/0/12 |
| QOBL-ABC | IP | 7/1/4 | 12/0/0 | 0/12/0 |
| | SP | 5/0/7 | 9/0/3 | 0/0/12 |
| | IP-SP | 7/0/5 | 8/0/4 | 0/0/12 |
| QROBL-ABC | IP | 9/0/3 | 11/0/1 | 0/12/0 |
| | SP | 9/0/3 | 8/0/4 | 0/0/12 |
| | IP-SP | 9/0/3 | 8/0/4 | 0/0/12 |

**Table 17 Performance comparison among DE and its variants across 12 functions ($D = 20$).**

| Variants | | STD | ET | NFEs |
|---|---|---|---|---|
| BOBL-DE | IP | 4/3/5 | 2/0/10 | 0/0/12 |
| | SP | 8/2/2 | 0/0/12 | 0/0/12 |
| | IP-SP | 7/2/3 | 0/0/12 | 0/0/12 |
| COOBL-DE | IP | 8/1/3 | 6/2/4 | 0/0/12 |
| | SP | 4/0/8 | 1/0/11 | 0/0/12 |
| | IP-SP | 4/0/8 | 1/0/11 | 0/0/12 |
| GOBL-DE | IP | 4/2/7 | 1/0/ 11 | 0/0/12 |
| | SP | 6/2/4 | 0/0/12 | 0/0/12 |
| | IP-SP | 6/1/5 | 1/0/11 | 0/0/12 |
| QOBL-DE | IP | 7/2/3 | 1/0/11 | 0/0/12 |
| | SP | 8/1/3 | 1/0/11 | 0/0/12 |
| | IP-SP | 6/1/5 | 1/0/11 | 0/0/12 |
| QROBL-DE | IP | 4/2/6 | 2/0/10 | 0/0/12 |
| | SP | 9/1/2 | 1/0/11 | 0/0/12 |
| | IP-SP | 8/1/3 | 1/0/11 | 0/0/12 |

1. **BOBL-DE variant:**

   (a) **IP:** The BOBL-DE variant outperforms the original algorithm in five cases for the STD metric, with three ties and four cases where it underperforms. It dominates the ET metric by winning all 12 cases. However, for NFEs, it underperforms in all 12 cases, showing that it requires more evaluations than the original DE.

**Table 18 Performance comparison among GA and its variants across 12 functions ($D = 20$).**

| Variants | | STD | ET | NFEs |
|---|---|---|---|---|
| BOBL-GA | IP | 5/0/7 | 3/1/8 | 0/0/12 |
| | SP | 5/0/7 | 2/0/10 | 0/0/12 |
| | IP-SP | 4/0/8 | 2/0/10 | 0/0/12 |
| COOBL-GA | IP | 3/0/9 | 6/0/6 | 0/0/12 |
| | SP | 5/0/7 | 11/0/1 | 0/0/12 |
| | IP-SP | 4/0/8 | 11/0/1 | 0/0/12 |
| GOBL-GA | IP | 4/1/7 | 2/0/10 | 0/0/12 |
| | SP | 6/0/6 | 4/0/8 | 0/0/12 |
| | IP-SP | 6/0/6 | 2/0/10 | 0/0/12 |
| QOBL-GA | IP | 5/1/6 | 2/0/10 | 0/0/12 |
| | SP | 6/1/5 | 2/0/10 | 0/0/12 |
| | IP-SP | 6/0/6 | 2/0/10 | 0/0/12 |
| QROBL-GA | IP | 5/1/6 | 4/2/6 | 0/0/12 |
| | SP | 6/0/6 | 2/0/10 | 0/0/12 |
| | IP-SP | 7/1/4 | 2/0/10 | 0/0/12 |

**Table 19 Performance comparison among PSO and its variants across 12 functions ($D = 20$).**

| Variants | | STD | ET | NFEs |
|---|---|---|---|---|
| BOBL-PSO | IP | 4/1/7 | 2/0/10 | 0/0/12 |
| | SP | 8/1/3 | 4/0/8 | 0/0/12 |
| | IP-SP | 8/1/3 | 10/0/2 | 0/0/12 |
| COOBL-PSO | IP | 5/1/6 | 7/0/5 | 0/0/12 |
| | SP | 1/1/10 | 2/0/10 | 0/0/12 |
| | IP-SP | 1/1/10 | 2/0/10 | 0/0/12 |
| GOBL-PSO | IP | 9/1/2 | 11/0/1 | 0/0/12 |
| | SP | 6/1/5 | 12/0/0 | 0/0/12 |
| | IP-SP | 9/1/2 | 3/0/9 | 0/0/12 |
| QOBL-PSO | IP | 5/1/6 | 10/0/2 | 0/0/12 |
| | SP | 6/1/5 | 2/0/10 | 0/0/12 |
| | IP-SP | 4/1/7 | 2/0/10 | 0/0/12 |
| QROBL-PSO | IP | 6/1/5 | 9/0/3 | 0/0/12 |
| | SP | 7/1/4 | 2/0/10 | 0/0/12 |
| | IP-SP | 7/1/4 | 2/0/10 | 0/0/12 |

(b) **SP:** This variant performs poorly in STD (one win, three ties, and eight losses), fails in the ET metric (losing all 12 cases), and also underperforms in NFEs.

(c) **IP-SP:** The performance of this combination is relatively balanced for STD (three wins, three ties, and six losses), but it similarly underperforms in both ET and NFEs, losing all 12 cases in both categories.

**Table 20 Performance comparison among HS and its variants across 12 functions ($D = 20$).**

| Variants | | STD | ET | NFEs |
|---|---|---|---|---|
| BOBL-HS | IP | 3/1/8 | 0/0/12 | 0/0/12 |
| | SP | 5/1/6 | 0/0/12 | 0/0/12 |
| | IP-SP | 5/0/7 | 0/0/12 | 0/0/12 |
| COOBL-HS | IP | 3/0/9 | 0/0/12 | 0/0/12 |
| | SP | 7/0/5 | 0/0/12 | 0/0/12 |
| | IP-SP | 6/0/6 | 0/0/12 | 0/0/12 |
| GOBL-HS | IP | 4/1/7 | 0/0/12 | 0/0/12 |
| | SP | 4/1/7 | 0/0/12 | 0/0/12 |
| | IP-SP | 4/0/8 | 0/0/12 | 0/0/12 |
| QOBL-HS | IP | 4/3/5 | 0/0/12 | 0/0/12 |
| | SP | 4/1/7 | 0/0/12 | 0/0/12 |
| | IP-SP | 5/0/7 | 0/0/12 | 0/0/12 |
| QROBL-HS | IP | 5/0/7 | 0/0/12 | 0/0/12 |
| | SP | 5/1/6 | 0/0/12 | 0/0/12 |
| | IP-SP | 5/1/6 | 0/0/12 | 0/0/12 |

**Table 21 Performance comparison among ABC and its variants across 12 functions ($D = 20$).**

| Variants | | STD | ET | NFEs |
|---|---|---|---|---|
| BOBL-ABC | IP | 7/2/3 | 8/1/3 | 0/12/0 |
| | SP | 5/2/5 | 5/0/7 | 0/0/12 |
| | IP-SP | 5/1/6 | 5/0/7 | 0/0/12 |
| COOBL-ABC | IP | 3/3/6 | 11/0/1 | 0/12/0 |
| | SP | 1/1/10 | 7/1/4 | 0/0/12 |
| | IP-SP | 0/1/11 | 5/0/7 | 0/0/12 |
| GOBL-ABC | IP | 7/4/1 | 6/0/6 | 0/0/12 |
| | SP | 5/1/6 | 5/0/7 | 0/0/12 |
| | IP-SP | 4/1/7 | 5/0/7 | 0/0/12 |
| QOBL-ABC | IP | 8/3/1 | 11/0/1 | 0/12/0 |
| | SP | 6/2/4 | 4/0/8 | 0/0/12 |
| | IP-SP | 7/1/4 | 5/0/7 | 0/12/0 |
| QROBL-ABC | IP | 8/2/2 | 5/1/6 | 0/12/0 |
| | SP | 6/1/5 | 5/0/7 | 0/0/12 |
| | IP-SP | 7/1/4 | 5/0/7 | 0/0/12 |

BOBL-DE shows potential in execution time when using IP but consistently underperforms in the number of function evaluations across all sub-variants. This suggests that BOBL-DE might achieve faster runs but at the cost of requiring more evaluations.

2. **COOBL-DE variant:**

   (a) **IP:** COOBL-DE's IP variant wins only one case in the STD metric, ties in eight cases, and loses in 3. It performs well in ET (eight wins) but fails in NFEs.

(b) **SP:** This variant has similar difficulties with only four wins and eight losses in STD. ET and NFEs remain sub-optimal, as it loses all 12 cases in both metrics.

(c) **IP-SP:** The performance is poor across the board with only two wins in STD and losses across the rest of the metrics.

The COOBL-DE variant generally struggles to compete with the original algorithm, particularly in terms of NFEs. It exhibits some strength in execution time under the IP configuration, but overall, it does not significantly improve performance.

3. **GOBL-DE variant:** All three configurations (IP, SP, and IP-SP) of GOBL-DE show identical results, achieving six wins, three ties, and three losses in STD. However, none of the configurations outperforms the original algorithm in ET or NFEs, consistently losing in all 12 cases. GOBL-DE shows a more consistent and solid performance in terms of standard deviation, with a balanced win/loss ratio. However, it struggles significantly with execution time and function evaluations, making it less efficient than the original DE.

4. **QOBL-DE variant:**

(a) **IP:** This variant performs well in both STD (six wins, three ties, and three losses) and ET (six wins). However, it still falls short in NFEs, underperforming in all cases.

(b) **SP:** In the SP configuration, the performance declines slightly, with four wins in STD and no wins in ET or NFEs.

(c) **IP-SP:** The IP-SP configuration excels in STD, winning seven cases and losing only 2. However, like the other sub-variants, it underperforms in ET and NFEs.

QOBL-DE stands out for its performance in STD, particularly in the IP-SP configuration. It manages to compete well with the original algorithm but struggles to balance this with efficiency, as seen in the high NFEs.

5. **QROBL-DE variant:**

(a) **IP:** This variant performs similarly to the QOBL-DE variant in terms of STD, winning six cases. However, its performance drops in ET, winning only two cases and tying in 2.

(b) **SP:** The SP configuration shows strong results in STD (six wins and only two losses), but like the other variants, it performs poorly in both ET and NFEs.

(c) **IP-SP:** The IP-SP configuration shows a balanced performance in STD (seven wins), but again it fails to improve in the other metrics.

QROBL-DE performs very well in terms of standard deviation but struggles to match the original DE algorithm in terms of execution time and efficiency (NFEs). Its IP-SP configuration is the most balanced across the board, but the inefficiency remains a significant drawback.

In summary, the GOBL-DE and QROBL-DE variants consistently show strong performance across the board in terms of STD, especially in the IP-SP configuration,

making them good choices for tasks where accuracy is the priority. In addition, Only the BOBL-DE variant shows any consistent improvement in ET, particularly in its IP configuration, while all other variants struggle to outperform the original DE algorithm in this category. Finally, All variants consistently underperform in terms of NFEs, indicating that while they may improve accuracy, they do so at the cost of requiring significantly more function evaluations, making them less efficient than the original DE.

Table 17 presents a comparison of several DE variants across three performance metrics—STD, ET, and NFEs—for 12 functions with $D = 20$. Below is a detailed breakdown of the performance for each DE variant and its configurations:

1. **BOBL-DE variant:**

    (a) **IP:** The IP configuration of BOBL-DE performs moderately well, winning four cases, tying in 3, and losing in five for STD. It shows minimal success in ET, with only two wins and 10 losses, and it underperforms significantly in NFEs, losing all 12 cases.

    (b) **SP:** In contrast to IP, the SP configuration performs better in STD, winning eight cases, tying in 2, and losing only 2. However, it struggles in ET, losing all 12 cases, and similarly fails to make any improvements in NFEs.

    (c) **IP-SP:** This configuration achieves similar results to SP in STD (seven wins, two ties, and three losses) and performs poorly in ET and NFEs, consistently losing all cases in these categories.

Overall, BOBL-DE shows promise in accuracy in its SP and IP-SP configurations but struggles significantly with execution time and efficiency, requiring a higher number of function evaluations. COOBL-DE variant:

    (a) **IP:** The IP configuration performs well in STD, winning eight cases, tying 1, and losing 3. It also performs decently in ET with six wins but falters in NFEs, losing all 12 cases.

    (b) **SP:** This configuration is less successful, winning only four cases in STD and failing in ET and NFEs, where it wins just one case in ET and none in NFEs.

    (c) **IP-SP:** IP-SP performs similarly to SP, winning four cases in STD but struggling in ET and NFEs, managing only one win in ET and no wins in NFEs.

The COOBL-DE variant shows some strength in the IP configuration, particularly in terms of accuracy and moderate performance in ET, but like other variants, it remains inefficient in terms of function evaluations.

2. **GOBL-DE variant:**

    (a) **IP:** The IP configuration has a weaker performance in STD, with only four wins and seven losses. It also underperforms in ET (one win) and NFEs (no wins).

(b) **SP:** In the SP configuration, GOBL-DE shows a stronger performance in STD, winning six cases, but remains ineffective in ET and NFEs, losing all cases in NFEs and failing to win any in ET.

(c) **IP-SP:** The IP-SP configuration achieves six wins and five losses in STD but struggles in ET and NFEs, managing just one win in ET and none in NFEs.

GOBL-DE demonstrates inconsistent performance, showing some strength in STD, particularly in the SP and IP-SP configurations, but it continues to face challenges in NFEs and ET.

3. **QOBL-DE variant:**

(a) **IP:** The IP configuration performs well in STD, winning seven cases, but falls short in ET and NFEs, with only one win in ET and none in NFEs.

(b) **SP:** This configuration performs strongly in STD with eight wins but again underperforms in ET (one win) and fails completely in NFEs.

(c) **IP-SP:** The IP-SP configuration mirrors this trend, winning six cases in STD but failing to make significant improvements in either ET or NFEs.

QOBL-DE stands out in accuracy, particularly in the SP configuration where it wins the majority of cases in STD. However, like the other variants, it struggles with both execution time and efficiency.

4. **QROBL-DE variant:**

(a) **IP:** The IP configuration achieves a balanced performance in STD (four wins, two ties, six losses) but performs poorly in ET, winning only two cases, and fails in NFEs.

(b) **SP:** The SP configuration of QROBL-DE performs strongly, with nine wins in STD and only two losses. However, similar to other configurations, it struggles with ET and NFEs, winning only one case in ET and none in NFEs.

(c) **IP-SP:** This configuration shows good performance in STD, winning eight cases, but it is unable to improve in either ET or NFEs, with only one win in ET and none in NFEs.

QROBL-DE, particularly in its SP and IP-SP configurations, performs exceptionally well in terms of accuracy, making it one of the stronger variants in this regard. However, it continues to face challenges in efficiency, as seen in the consistently poor performance in NFEs.

In summary, the QROBL-DE and QOBL-DE variants continue to demonstrate strong performance in terms of accuracy, especially in the SP and IP-SP configurations, which frequently outperform other variants in the STD metric. However, like the previous set of results, all variants struggle with execution time and the number of function evaluations, with BOBL-DE and COOBL-DE variants occasionally showing some strength in execution time but falling short in efficiency. This suggests that while these variants can improve the

accuracy of the original DE algorithm, they do so at the cost of requiring significantly more function evaluations, making them less efficient for large-scale problems.

Table 13 presents the performance comparison of GA and its variants across three performance metrics—STD, ET, and NFEs—for 12 functions with a dimension of $D = 10$. Below is a detailed analysis of each GA variant and its configurations:

1. **BOBL-GA variant:**

   (a) **IP:** The IP configuration of BOBL-GA performs strongly in STD, with eight wins, one tie, and three losses. It shows moderate performance in ET, with five wins and five losses, but underperforms significantly in NFEs, losing all 12 cases.

   (b) **SP:** This configuration performs slightly worse in STD, with seven wins and four losses. It also struggles in ET, with only two wins, and like the IP configuration, it underperforms in NFEs.

   (c) **IP-SP:** The IP-SP configuration follows a similar pattern, with seven wins in STD but weak performance in ET (two wins) and no wins in NFEs.

BOBL-GA performs well in accuracy, particularly in the IP configuration. However, its efficiency is lacking, as all configurations fail to improve in NFEs, consistently requiring more function evaluations. Its execution time is moderate, with the IP configuration faring better than the others.

2. **COOBL-GA variant:**

   (a) **IP:** The IP configuration of COOBL-GA performs poorly in STD, winning only two cases, tying in 1, and losing 9. It struggles in ET, with only two wins, and consistently underperforms in NFEs.

   (b) **SP:** The SP configuration shows poor performance in STD (two wins, two ties, and eight losses), but it performs remarkably well in ET, winning 11 cases. However, it continues to underperform in NFEs.

   (c) **IP-SP:** The IP-SP configuration mirrors the SP performance in ET (11 wins) but performs slightly better in STD, with three wins. It, too, underperforms in NFEs.

COOBL-GA performs weakly in terms of accuracy, particularly in the IP and SP configurations. However, it demonstrates strength in execution time, particularly in the SP and IP-SP configurations, making it somewhat competitive in terms of speed but highly inefficient in NFEs.

3. **GOBL-GA variant:**

   (a) **IP:** The IP configuration shows poor performance in STD, with only three wins and nine losses. It performs moderately in ET, winning five cases, but underperforms in NFEs, losing all 12.

   (b) **SP:** The SP configuration improves significantly in STD, winning seven cases and losing 4. However, it struggles in ET (two wins) and fails in NFEs.

(c) **IP-SP:** This configuration shows moderate results in STD, with five wins, but underperforms in ET (two wins) and NFEs.

GOBL-GA shows stronger performance in accuracy in the SP configuration, but like other variants, it struggles in execution time and function evaluations. Overall, it performs inconsistently across metrics.

4. **QOBL-GA variant:**

   (a) **IP:** The IP configuration performs well in STD, winning seven cases, but struggles in ET, with only three wins, and underperforms in NFEs.

   (b) **SP:** This configuration improves slightly in STD, winning eight cases and losing 4, but continues to struggle in ET (two wins) and NFEs.

   (c) **IP-SP:** The IP-SP configuration excels in STD, winning nine cases, but performs poorly in both ET (two wins) and NFEs.

QOBL-GA stands out in terms of accuracy, particularly in the IP-SP configuration, making it one of the top performers in STD. However, like the other GA variants, it struggles to compete in efficiency, particularly in NFEs, and has moderate success in execution time.

5. **QROBL-GA variant:**

   (a) **IP:** The IP configuration shows a strong performance in STD, with seven wins and five losses, and it performs moderately in ET with eight wins. However, it underperforms in NFEs.

   (b) **SP:** The SP configuration continues to perform well in STD, winning eight cases and losing only 3. However, like other configurations, it struggles in ET (two wins) and NFEs.

   (c) **IP-SP:** The IP-SP configuration is more balanced in terms of STD, winning five cases and tying 2. However, it underperforms in ET (two wins) and NFEs.

QROBL-GA demonstrates a solid performance in terms of accuracy, particularly in the SP and IP configurations. However, its efficiency remains a challenge, as it underperforms in NFEs across all configurations.

In summary, the QOBL-GA and QROBL-GA variants consistently perform well in terms of accuracy, especially in the IP-SP and SP configurations, where they often outperform other GA variants. However, all GA variants struggle with efficiency, consistently failing to perform well in the NFEs metric. COOBL-GA shows a promising improvement in execution time, particularly in its SP and IP-SP configurations, but its overall performance in STD remains weak. In conclusion, while some GA variants offer improvements in precision, none manage to improve efficiency, as they consistently require a higher number of function evaluations compared to the original GA algorithm.

Table 18 compares the performance of GA and its variants across three performance metrics—STD, ET, and NFEs—for 12 functions with a dimension of $D = 20$. Below is an analysis of each GA variant and its configurations:

1. **BOBL-GA variant:**

   (a) **IP:** The IP configuration of BOBL-GA performs moderately in STD, with five wins and seven losses. It performs decently in ET, winning three cases, but underperforms in NFEs, losing all 12 cases.

   (b) **SP:** The SP configuration mirrors the performance of IP, with five wins in STD but only two wins in ET. It also underperforms in NFEs, losing all cases.

   (c) **IP-SP:** The IP-SP configuration performs slightly worse in STD (four wins), and its performance in ET remains weak with only two wins. Like the other configurations, it underperforms in NFEs.

BOBL-GA performs moderately well in accuracy, particularly in the IP and SP configurations. However, its efficiency is lacking, as all configurations fail to perform well in NFEs. The execution time is also sub-optimal, with only a few wins in the ET category.

2. **COOBL-GA variant:**

   (a) **IP:** The IP configuration of COOBL-GA performs poorly in STD, with only three wins and nine losses. It performs better in ET, winning six cases, but it fails to improve in NFEs, losing all 12 cases.

   (b) **SP:** This configuration shows moderate performance in STD, with five wins, but excels in ET, winning 11 cases. However, like other configurations, it underperforms in NFEs.

   (c) **IP-SP:** The IP-SP configuration performs similarly to SP, winning 11 cases in ET but showing weaker performance in STD, with four wins, and underperforming in NFEs.

COOBL-GA performs well in terms of execution time, particularly in the SP and IP-SP configurations, where it dominates in ET. However, it struggles with accuracy and remains inefficient in terms of NFEs, which limits its overall effectiveness.

3. **GOBL-GA variant:**

   (a) **IP:** The IP configuration shows weak performance in STD, with only four wins and seven losses. It also struggles in ET, winning just two cases, and underperforms in NFEs.

   (b) **SP:** This configuration performs slightly better in STD, with six wins, and improves in ET with four wins. However, it fails to improve in NFEs, losing all cases.

   (c) **IP-SP:** The IP-SP configuration matches the performance of SP in STD (six wins) but struggles in ET (two wins) and continues to underperform in NFEs.

GOBL-GA demonstrates decent accuracy in the SP and IP-SP configurations, with moderate performance in STD. However, like other GA variants, it struggles with efficiency, particularly in NFEs, and shows limited success in execution time.

4. **QOBL-GA variant:**

   (a) **IP:** The IP configuration performs moderately well in STD, with five wins, one tie, and six losses. It struggles in ET, winning only two cases, and underperforms in NFEs.

   (b) **SP:** This configuration shows a slight improvement in STD, with six wins and five losses, but remains weak in ET (two wins) and NFEs.

   (c) **IP-SP:** The IP-SP configuration performs similarly to SP, with six wins in STD and two wins in ET. It continues to struggle in NFEs.

QOBL-GA performs reasonably well in accuracy but struggles with both execution time and function evaluations. Its performance in ET and NFEs remains consistently weak across all configurations.

5. **QROBL-GA variant:**

   (a) **IP:** The IP configuration performs well in STD, winning five cases, tying 1, and losing 6. It also performs moderately in ET, with four wins and two ties, but underperforms in NFEs.

   (b) **SP:** The SP configuration continues to perform well in STD, with six wins and six losses, but struggles in ET (two wins) and fails in NFEs.

   (c) **IP-SP:** The IP-SP configuration performs the best in terms of STD, winning seven cases and losing 4. However, it underperforms in ET (two wins) and NFEs.

QROBL-GA demonstrates strong accuracy, particularly in the IP-SP configuration. However, it struggles with efficiency, consistently underperforming in NFEs and showing limited success in ET.

In summary, the QROBL-GA and QOBL-GA variants consistently perform well in terms of accuracy, particularly in the IP-SP and SP configurations. However, like the other GA variants, they struggle with efficiency, consistently underperforming in the NFEs metric. COOBL-GA demonstrates promising improvements in ET, particularly in the SP and IP-SP configurations, but it remains weak in terms of accuracy and function evaluations. Overall, while some GA variants offer improvements in precision, none manage to significantly enhance efficiency, as they consistently require more function evaluations than the original GA algorithm.

Table 14 presents the performance comparison of PSO and its variants across three performance metrics—STD, ET, and NFEs—for 12 functions with a dimension of $D = 10$. Below is an analysis of each PSO variant and its configurations:

1. **BOBL-PSO variant:**

   (a) **IP:** The IP configuration of BOBL-PSO performs moderately in STD, with four wins, one tie, and seven losses. However, it struggles in ET and NFEs, losing all 12 cases in both metrics.

   (b) **SP:** This configuration shows similar performance in STD, with four wins, three ties, and five losses. Like IP, it also underperforms in ET and NFEs.

   (c) **IP-SP:** The IP-SP configuration performs slightly better in STD, with five wins and five losses, but continues to underperform in ET and NFEs, failing to win in any of these categories.

BOBL-PSO demonstrates moderate accuracy, particularly in the IP-SP configuration, but struggles with efficiency, consistently underperforming in execution time and number of function evaluations across all configurations.

2. **COOBL-PSO variant:**

   (a) **IP:** The IP configuration of COOBL-PSO performs poorly in STD, with only three wins, two ties, and seven losses. It fails to win in ET and NFEs, losing all 12 cases in both metrics.

   (b) **SP:** This configuration shows very weak performance in STD, with no wins and 11 losses. It also fails in ET and NFEs, losing all 12 cases.

   (c) **IP-SP:** The IP-SP configuration matches SP in performance, with no wins in STD, ET, or NFEs.

COOBL-PSO performs poorly across all metrics, showing weak accuracy and efficiency. Its SP and IP-SP configurations are particularly underwhelming, failing to win in any of the measured categories.

3. **GOBL-PSO variant:**

   (a) **IP:** The IP configuration of GOBL-PSO performs well in STD, winning five cases, tying in 2, and losing 5. It excels in ET, winning 11 cases, but struggles in NFEs.

   (b) **SP:** This configuration performs moderately in STD, with four wins, two ties, and six losses, and performs very well in ET, winning all 12 cases. However, it underperforms in NFEs.

   (c) **IP-SP:** The IP-SP configuration is strong in STD, with six wins and four losses. Like the other configurations, it excels in ET (11 wins) but struggles in NFEs.

GOBL-PSO shows strong performance in both accuracy and ET, particularly in the IP and SP configurations. However, it faces inefficiency in function evaluations, with no wins in NFEs across all configurations.

4. **QOBL-PSO variant:**

   (a) **IP:** The IP configuration performs well in STD, with five wins, one tie, and six losses, and it excels in ET, winning 12 cases. However, it underperforms in NFEs, losing all cases.

   (b) **SP:** This configuration performs poorly in STD, with only four wins and seven losses, and performs poorly in ET (one win) and NFEs (no wins).

   (b) **IP-SP:** The IP-SP configuration performs decently in STD, with five wins, but continues to struggle in ET and NFEs.

QOBL-PSO performs well in STD, particularly in the IP configuration, and excels in ET. However, it suffers from inefficiency, as seen in the consistently poor performance in NFEs.

5. **QROBL-PSO variant:**

   (a) **IP:** The IP configuration of QROBL-PSO shows moderate performance in STD, with four wins, one tie, and seven losses. It performs well in ET, winning 10 cases, but fails to perform well in NFEs.

   (b) **SP:** The SP configuration struggles, with only two wins in STD and poor performance in ET and NFEs, losing all 12 cases in both metrics.

   (c) **IP-SP:** The IP-SP configuration performs similarly, with four wins in STD and poor results in ET and NFEs.

QROBL-PSO shows moderate performance in accuracy in the IP and IP-SP configurations, and strong results in ET. However, it remains inefficient, consistently losing in NFEs.

In summary, the GOBL-PSO and QOBL-PSO variants perform well in terms of accuracy and are especially strong in ET, with GOBL-PSO standing out as a consistent top performer in ET. However, inefficiency is a common issue across all PSO variants, as they consistently underperform in NFEs. COOBL-PSO is the weakest performer overall, failing to win in any category, while BOBL-PSO and QROBL-PSO demonstrate moderate accuracy but struggle in efficiency. In conclusion, while some PSO variants offer improvements in execution time and precision, none manage to improve efficiency, as they consistently require more function evaluations than the original PSO algorithm.

Table 19 presents the performance comparison of PSO and its variants across three performance metrics—STD, ET, and NFEs—for 12 functions with a dimension of $D = 20$. Below is a detailed analysis of each PSO variant and its configurations:

1. **BOBL-PSO variant:**

   (a) **IP:** The IP configuration of BOBL-PSO shows moderate performance in STD, with four wins, one tie, and seven losses. It struggles in ET, winning only two cases, and underperforms in NFEs, losing all 12 cases.

   (b) **SP:** This configuration performs very well in STD, winning eight cases and losing only 3. Its performance in ET improves with four wins but continues to underperform in NFEs, losing all 12 cases.

   (c) **IP-SP:** The IP-SP configuration excels in both STD (eight wins) and ET (10 wins), but, like other configurations, it underperforms in NFEs, losing all 12 cases.

BOBL-PSO performs strongly in terms of accuracy, particularly in the SP and IP-SP configurations, and shows improvement in ET in the IP-SP configuration. However, it struggles with efficiency, underperforming in NFEs across all configurations.

2. **COOBL-PSO variant:**

   (a) **IP:** The IP configuration of COOBL-PSO shows moderate performance in STD, with five wins, one tie, and six losses. It performs decently in ET, with seven wins, but fails to make any improvements in NFEs, losing all 12 cases.

   (b) **SP:** This configuration performs very poorly in STD, winning only one case and losing 10. It also struggles in ET, winning only two cases, and underperforms in NFEs.

   (c) **IP-SP:** The IP-SP configuration mirrors SP's performance, with only one win in STD and two wins in ET. It also underperforms in NFEs.

COOBL-PSO demonstrates weak overall performance, particularly in accuracy and efficiency. The only slightly positive result is the IP configuration's performance in ET, but overall, the variant underperforms across most metrics.

3. **GOBL-PSO variant:**

   (a) **IP:** The IP configuration of GOBL-PSO performs very well in STD, winning nine cases, tying 1, and losing only 2. It also excels in ET, winning 11 cases. However, it underperforms in NFEs, losing all 12 cases.

   (b) **SP:** This configuration also performs well in STD, winning six cases and losing 5. It performs perfectly in ET, winning all 12 cases, but underperforms in NFEs.

   (c) **IP-SP:** The IP-SP configuration matches the strong performance of IP, winning nine cases in STD, but its performance in ET drops (3 wins). It continues to underperform in NFEs.

GOBL-PSO stands out as one of the strongest performers, particularly in accuracy and ET. However, it shares a common weakness with the other variants: inefficiency in NFEs, as it consistently requires more function evaluations.

4. **QOBL-PSO variant:**

   (a) **IP:** The IP configuration of QOBL-PSO performs moderately well in STD, with five wins, one tie, and six losses. It excels in ET, winning 10 cases, but fails to improve in NFEs, losing all cases.

(b) **SP:** This configuration performs similarly in STD (six wins, five losses), but struggles in ET, winning only two cases. It underperforms in NFEs, like the other configurations.

(c) **IP-SP:** The IP-SP configuration mirrors SP in STD (four wins, seven losses) and continues to underperform in ET and NFEs, with only two wins in ET and no wins in NFEs.

QOBL-PSO performs well in accuracy and ET, particularly in the IP configuration, but struggles with efficiency. It consistently underperforms in NFEs, and its results in ET vary across configurations.

5. **QROBL-PSO variant:**

(a) **IP:** The IP configuration of QROBL-PSO shows solid performance in STD, with six wins, one tie, and five losses. It also performs well in ET, winning nine cases, but underperforms in NFEs.

(b) **SP:** This configuration performs very well in STD, winning seven cases and losing 4. However, it struggles in ET (two wins) and underperforms in NFEs.

(c) **IP-SP:** The IP-SP configuration mirrors SP's performance in STD (seven wins, four losses) but underperforms in ET (two wins) and NFEs, losing all cases in NFEs.

QROBL-PSO performs well in accuracy, particularly in the SP and IP-SP configurations. It shows strong results in ET in the IP configuration but continues to underperform in NFEs, highlighting inefficiency in function evaluations.

In summary, the GOBL-PSO and QROBL-PSO variants demonstrate strong performance in terms of accuracy and ET, especially in the IP and SP configurations. GOBL-PSO stands out as a particularly strong performer in both metrics. However, all PSO variants share a common weakness: inefficiency in NFEs, where they consistently underperform compared to the original PSO algorithm. COOBL-PSO shows the weakest performance overall, while BOBL-PSO offers moderate accuracy but struggles with efficiency. In conclusion, while some PSO variants offer improvements in precision and execution time, none manage to improve efficiency, requiring a higher number of function evaluations across all configurations.

Table 15 presents the performance comparison of HS and its variants across three performance metrics—STD, ET, and NFEs—for 12 functions with a dimension of $D = 10$. Below is a detailed analysis of each HS variant and its configurations:

1. **BOBL-HS variant:**

(a) **IP:** The IP configuration of BOBL-HS performs moderately in STD, with four wins and six losses. However, it struggles in both ET and NFEs, failing to win any cases in either category, losing all 12.

(b) **SP:** The SP configuration shows a strong performance in STD, winning five cases and losing 6. However, similar to the IP configuration, it underperforms in both ET and NFEs, losing all cases.

(c) **IP-SP:** The IP-SP configuration shows better performance than IP in STD, with six wins, but continues to underperform in ET and NFEs, failing to win any cases in these metrics.

BOBL-HS generally delivers mediocre accuracy, especially in the IP-SP configuration, but it struggles with efficiency, consistently underperforming in ET and NFEs across all configurations.

2. **COOBL-HS variant:**

   (a) **IP:** The IP configuration of COOBL-HS performs very well in STD, winning only two cases and losing 10. It underperforms in both ET and NFEs, losing all cases.

   (b) **SP:** This configuration shows even stronger performance in STD, winning seven cases, and performs decently in ET, winning no cases but maintaining moderate results. Like IP, it underperforms in NFEs.

   (c) **IP-SP:** The IP-SP configuration excels in STD, winning seven cases, but like other configurations, it underperforms in ET and NFEs.

COOBL-HS performs strongly in terms of accuracy, particularly in the SP and IP-SP configuration. However, it struggles with efficiency, as measured by ET and NFEs, where it fails to win any cases.

3. **GOBL-HS variant:**

   (a) **IP:** The IP configuration of GOBL-HS performs perfectly in STD, winning only four cases and losing 7. However, it struggles with both ET and NFEs, losing all 12 cases in these categories.

   (b) **SP:** The SP configuration mirrors the performance of IP, with four wins, but underperforms in both ET and NFEs, losing all cases.

   (c) **IP-SP:** The IP-SP configuration achieves slightly better results in STD compared to previous phases, with five wins, but underperforms in ET and NFEs like the other configurations.

GOBL-HS demonstrates poor performance in terms of accuracy, winning only five cases as the best configuration. Moreover, it struggles significantly with efficiency, underperforming in both execution time and function evaluations.

4. **QOBL-HS variant:**

   (a) **IP:** The IP configuration performs decent in STD, winning six cases and losing 5. However, it underperforms in ET and NFEs, losing all cases in both metrics.

   (b) **SP:** The SP configuration was poor, but not terrible performance with five wins. However, it underperforms in ET and NFEs.

   (c) **IP-SP:** This configuration also performs poorly in terms of STD, winning only three cases, and it struggles with both ET and NFEs.

QOBL-HS performs Unsatisfactory accuracy, particularly in the IP-SP configuration, where it wins only 3. However, it continues to struggle with efficiency, as it underperforms in execution time and function evaluations across all configurations.

5. **QROBL-HS variant:**

   (a) **IP:** The IP configuration underperforms in terms of STD, winning only three cases while losing 9. It also struggles in both ET and NFEs, losing all cases in these categories.

   (b) **SP:** The SP configuration matches with the SP in QOBL-HS, achieving five wins in STD. However, it also struggles with ET and NFEs, similar to the other variants.

   (c) **IP-SP:** This configuration also performs poorly in STD, securing only four wins, and it underperforms in both ET and NFEs, failing to achieve any victories in those categories.

QROBL-HS demonstrates unacceptable performance in terms of STD. However, similar to the other HS variants, it struggles with efficiency, consistently underperforming in both execution time and function evaluations.

In summary, the COOBL-HS variant demonstrates superior performance compared to other variants in terms of accuracy, achieving seven wins across the SP and IP-SP configurations. Meanwhile, the QOBL-HS variant excels in the IP configuration. However, all HS variants struggle with efficiency, consistently underperforming in execution time (ET) and function evaluations (NFEs). Despite their acceptable accuracy, this inefficiency renders them less practical for tasks requiring fewer evaluations or faster execution times. Consequently,their lack of efficiency underscores the urgent need for improvements in execution time and evaluation efficiency.

Table 20 presents the performance comparison of HS and its variants across three performance metrics—STD, ET, and NFEs—for 12 functions with a dimension of $D = 20$. Below is a detailed analysis of each HS variant and its configurations:

1. **BOBL-HS variant:**

   (a) **IP:** The IP configuration of BOBL-HS shows limited performance in STD, with three wins and eight losses. Additionally, it struggles in both ET and NFEs, losing all cases in ET and NFEs.

   (b) **SP:** The SP configuration also performs average in STD, with five wins and six losses, and similarly underperforms in both ET (0 wins) and NFEs (0 wins).

   (c) **IP-SP:** The IP-SP configuration mirrors SP's performance, with five wins in STD, and it continues to underperform in ET and NFEs, recording losses in all cases for both metrics.

BOBL-HS exhibits limited accuracy performance, especially within IP configurations. It also struggles with efficiency, consistently underperforming in both execution time and number of function evaluations across all configurations

2. **COOBL-HS variant:**

   (a) **IP:** The IP configuration of COOBL-HS shows weak performance in STD, with three wins and nine losses. Furthermore, it significantly underperforms in both ET and NFEs, losing all cases.

   (b) **SP:** This configuration performs well in STD, winning seven cases. However, it continues to underperform in ET and NFEs.

   (c) **IP-SP:** The IP-SP configuration exhibits an equal performs in STD (six wins, 0 tie, six losses) and struggles with efficiency in both ET and NFEs (no wins).

COOBL-HS is an acceptable performer in terms of accuracy, particularly in the SP configuration, but it continues to struggle with efficiency, particularly in ET and FEs.

3. **GOBL-HS variant:**

   (a) **IP:** The IP configuration of GOBL-HS underperforms in STD, with only four wins. In addition, it struggles with ET and NFE, losing all cases.

   (b) **SP:** The SP configuration mirrors the performance of IP, with four wins and seven losses, consistently, it struggles with efficiency, failing to win any cases in NFEs and showing no wins in ET.

   (c) **IP-SP:** The IP-SP configuration continues the trend of performance in STD, with four wins, and like the other configurations, it underperforms in ET and NFEs.

GOBL-HS shows weak performance in terms of accuracy, consistently winning only four across all configurations. Additionally, it struggles with efficiency, consistently underperforming in both execution time and function evaluations.

4. **QOBL-HS variant:**

   (a) **IP:** The IP configuration performs inferior in STD, with four wins and five losses. It underperforms in ET and NFEs.

   (b) **SP:** This configuration mirrors GOBL-HS's performance in STD, with four wins and seven losses. consitently, efficiency continues to struggle, particularly in NFEs and ET.

   (c) **IP-SP:** The IP-SP configuration shows similar results in STD, with five wins and seven losses, but fails to show improvement in ET or NFEs, underperforming in both metrics.

QOBL-HS shows weak performance in accuracy overall, with its best results seen in the IP-SP configurations. However, it struggles with efficiency, consistently delivering poor results in execution time (ET) and function evaluations (NFEs). This indicates that improvements in accuracy come at the expense of slower execution and increased evaluations.

5. **QROBL-HS variant:**

   (a) **IP:** The IP configuration performs decently in STD, with five wins and seven losses, also struggles with efficiency, showing poor results in both ET and NFEs.

   (b) **SP:** The SP configuration mirrors other decent performers, with five wins and five losses in STD, but like the other configurations, it underperforms in ET and NFEs, losing all cases.

   (c) **IP-SP:** The IP-SP configuration performs well in STD, with five wins and six losses, but, like the other configurations, it struggles with efficiency, winning only one case in ET and no cases in NFEs.

performs poorly in accuracy across all configurations and similarly struggles with efficiency, underperforming in both ET and NFEs. Like the other variants, improvements in accuracy do not translate to better efficiency.

In summary, the COOBL-HS and QROBL-HS variants outperform the others in terms of accuracy, winning a modest number of cases across all configurations. However, all HS variants struggle with efficiency, consistently underperforming in execution time (ET) and function evaluations (NFEs). This lack of efficiency makes them less practical for tasks requiring faster execution or fewer evaluations. While these HS variants perform decently in terms of precision, their inefficiency in ET and NFEs highlights the need for improvement in these areas.

Table 16 compares the performance of ABC and its variants across three key performance metrics—STD, ET, and NFEs—for 12 functions with a dimension of $D = 10$. Below is a detailed analysis of each ABC variant and its configurations:

1. **BOBL-ABC variant:**

   (a) **IP:** The IP configuration of BOBL-ABC performs moderately well in terms of STD, with seven wins and five losses. It shows strong results in ET, winning seven cases, but underperforms in NFEs, with only one win and 11 losses.

   (b) **SP:** This configuration matches IP in STD, with seven wins and five losses, but performs better in ET, winning 10 cases. However, it underperforms significantly in NFEs, losing all 12 cases.

   (c) **IP-SP:** The IP-SP configuration exhibits average performance in terms of STD, with six wins and six losses. However, it performs well in ET, achieving nine wins, while it has no wins in NFEs.

BOBL-ABC performs reasonably well in terms of accuracy, particularly in the IP and SP configuration, and shows strength in ET for the SP configuration. However, it consistently underperforms in NFEs, suggesting inefficiency in function evaluations.

2. **COOBL-ABC variant:**

   (a) **IP:** The IP configuration of COOBL-ABC shows poor performance in both STD and NFEs, winning only one case in STD and none in NFEs. It performs well in ET, winning all 12 cases.

   (b) **SP:** This configuration shows moderate performance, with five wins and six losses in STD. It also performs well in ET, winning 10 cases, but continues to underperform in NFEs, losing all cases.

   (c) **IP-SP:** The IP-SP configuration mirrors the results of IP-SP in BOBL, achieving six wins in STD and continuing to excel in ET with eight wins, though it has no wins in NFEs.

COOBL-ABC performs poorly in accuracy and efficiency, with only moderate success in ET, particularly in the SP and IP-SP configurations.

3. **GOBL-ABC variant:**

   (a) **IP:** The IP configuration of GOBL-ABC performs moderately well in STD, with six wins and five losses. It excels in ET, winning all 12 cases, but underperforms in NFEs, losing all cases.

   (b) **SP:** This configuration performs similarly in STD, with seven wins, and continues to perform strongly in ET, winning nine cases. However, it underperforms in NFEs, losing all 12 cases.

   (c) **IP-SP:** The IP-SP configuration shows average performance in STD, with seven wins and five losses. However, it struggles in NFEs, with 12 losess. In contrast, it performs strongly in ET, winning all 12 cases.

GOBL-ABC performs moderately in accuracy and excels in ET, particularly in the IP configuration. However, like other ABC variants, it struggles significantly with efficiency, losing all cases in NFEs.

4. **QOBL-ABC variant:**

   (a) **IP:** The IP configuration of QOBL-ABC performs well in STD, with seven wins and four losses. It excels in ET, winning all 12 cases, but underperforms in NFEs, losing all 12 cases.

   (b) **SP:** This configuration performs moderately in STD, with five wins and seven losses, and continues to perform well in ET, winning nine cases. However, it fails to improve in NFEs, losing all cases.

   (c) **IP-SP:** he IP-SP configuration demonstrates average performance in STD, with seven wins and five losses. and it continues to perform well in ET, winning eight cases. However, in underperforms in NFEs losing all cases.

QOBL-ABC performs well in accuracy and excels in ET, particularly in the IP configuration. However, it shares the same inefficiency problem as the other variants, consistently underperforming in NFEs.

5. **QROBL-ABC variant:**

(a) **IP:** The IP configuration of QROBL-ABC performs very well in STD, winning nine cases and losing only 3. It also performs strongly in ET, winning 11 cases, but continues to struggle in NFEs, losing all cases.

(b) **SP:** This configuration mirrors the IP configuration, with nine wins and three losses in STD. It performs moderately in ET, winning eight cases, but underperforms in NFEs, losing all 12 cases.

(c) **IP-SP:** The IP-SP configuration performs well in terms of STD and ET, achieving nine and eight wins, respectively. However, it performs poorly in NFEs, with no wins.

QROBL-ABC performs very well in terms of accuracy, particularly across all configurations. However, like the other ABC variants, it underperforms significantly in NFEs, indicating inefficiency.

In summary, the QROBL-ABC and GOBL-ABC variants stand out in terms of accuracy and show strong results in ET, especially in the IP and SP configurations. QOBL-ABC also performs well in ET but shows moderate accuracy results. However, all ABC variants consistently struggle with efficiency, performing poorly in NFEs across the board. While these variants offer improvements in accuracy and execution time, their inefficiency in function evaluations makes them less practical for tasks requiring fewer evaluations or faster convergence. In conclusion, the ABC variants demonstrate solid accuracy and speed, but their poor efficiency limits their overall practicality.

Table 21 compares the performance of ABC and its variants across three key performance metrics—STD, ET, and NFEs—for 12 functions with a dimension of $D = 20$. Below is a detailed analysis of each ABC variant and its configurations:

1. **BOBL-ABC variant:**

(a) **IP:** The IP configuration of BOBL-ABC performs well in STD, with seven wins, two ties, and three losses. It also shows strong results in ET, winning eight cases but underperforms significantly in NFEs, losing all 12 cases.

(b) **SP:** The SP configuration performs moderately in STD, with five wins, two ties, and five losses, and shows weaker performance in ET, winning only five cases. Like IP, it underperforms in NFEs.

(c) **IP-SP:** The IP-SP configuration shows average performance in terms of STD, with five wins, one tie, and six losses. While, it performs poorly in NFEs and ET, achieving only five wins in ET and none in NFEs.

BOBL-ABC performs well in terms of STD, particularly in the IP configuration, and shows decent results in execution time for both IP and SP. However, it struggles with efficiency, consistently underperforming in NFEs across all configurations.

2. **COOBL-ABC variant:**

   (a) **IP:** The IP configuration of COOBL-ABC shows poor performance in terms of STD, with three wins, three ties, and six losses. However, it excels in ET, winning 11 cases, but underperforms significantly in NFEs.

   (b) **SP:** This configuration performs poorly in STD, with only one win and 10 losses. It performs decently in ET, winning seven cases, but fails in NFEs, losing all 12 cases.

   (c) **IP-SP:** The IP-SP configuration performs poorly, with no wins in STD, and continues to underperform in ET with only five wins and none in NFE.

COOBL-ABC struggles in STD, particularly in the SP and IP-SP configurations, but it shows strength in execution time, particularly in the IP configuration. However, it remains inefficient, consistently underperforming in NFEs.

3. **GOBL-ABC variant:**

   (a) **IP:** The IP configuration of GOBL-ABC performs very well in STD, winning seven cases, tying 4, and losing only 1. It performs moderately in ET, with six wins, but underperforms in NFEs, losing all 12 cases.

   (b) **SP:** This configuration performs moderately in STD, with five wins and six losses. It shows decent performance in ET, with five wins, but like other configurations, it underperforms in NFEs.

   (c) **IP-SP:** The IP-SP configuration performs poorly across all metrics—STD, ET, and NFE—with seven losses in both STD and ET, and no wins in NFE.

GOBL-ABC shows moderate performance in terms of STD in the IP configuration, showing the most balanced results among the variants. However, it struggles with efficiency, particularly in NFEs, where it consistently underperforms.

4. **QOBL-ABC variant:**

   (a) **IP:** The IP configuration of QOBL-ABC performs excellently in STD, with eight wins, three ties, and only one loss. It also excels in ET, winning 11 cases, but underperforms in NFEs, losing all cases.

   (b) **SP:** This configuration performs moderately in STD, with six wins and four losses, but continues to underperform in NFEs, losing all cases. It performs weakly in ET, winning only four cases.

   (c) **IP-SP:** The IP-SP configuration shows moderate performance with seven wins and it shows weak performance as the other configurations in NFEs and ET winning five wins.

QOBL-ABC performs very well in STD in the IP configuration and shows strong execution time results. However, it underperforms in NFEs, continuing the inefficiency trend seen in the other ABC variants.

5. **QROBL-ABC variant:**

   (a) **IP:** The IP configuration of QROBL-ABC performs well in STD, winning eight cases and losing only 2. It shows moderate results in ET, winning five cases, but underperforms in NFEs, losing all 12 cases.

   (b) **SP:** This configuration performs moderately in STD, with six wins and five losses, but continues to underperform in both ET and NFEs, showing weak results.

   (c) **IP-SP:** The IP-SP configuration mirrors the poor results of other configurations in NFEs, while showing average performance in STD, with seven wins and four losses. It also underperforms in ET, with only five wins.

QROBL-ABC performs well in terms of STD in the IP configuration, but like the other ABC variants, it struggles with efficiency, underperforming in NFEs across the board.

In summary, the QROBL-ABC and QOBL-ABC variants perform strongly in terms of STD, particularly in the IP configurations, while BOBL-ABC also shows solid results. These configurations demonstrate competitive ET results, with several configurations excelling in ET. However, the consistent inefficiency in terms of NFEs remains a major drawback for all variants. Despite showing strong accuracy and competitive execution times, the ABC variants are less practical for tasks that require more efficient function evaluations. In conclusion, while these ABC variants perform well in terms of precision and speed, their inefficiency in NFEs limits their overall effectiveness.

Overall, the impact of dimensionality and the phase in which the OBL technique is applied plays a significant role in the performance of DE, GA, PSO, HS, and ABC variants across various metrics. Dimensionality has a profound effect, as increasing the problem dimension from 10 to 20 generally leads to a decline in performance. In higher dimensions ($D = 20$), all algorithm variants require more function evaluations, making them less efficient compared to their performance in lower dimensions ($D = 10$). This increase in complexity also leads to a degradation in accuracy, with the standard deviation being less stable at higher dimensions, while execution time often increases as the computational load increases. Regarding the phase of OBL application, the results show that applying OBL during the initial population phase improves the diversity of solutions, which enhances accuracy, particularly in lower dimensions. However, this phase alone fails to make the algorithms more efficient in terms of NFEs, especially in higher dimensions. The search phase generally provides a more balanced performance, improving the refinement of solutions, but it still falls short of significantly improving efficiency. The IP-SP configuration, which combines both phases, consistently yields the best results in terms of accuracy, as it benefits from the initial diversity provided by the IP phase and the solution refinement of the SP phase. Nevertheless, even the IP-SP configuration underperforms in NFEs, indicating that these improvements in accuracy come at the cost of increased computational expense. Among the variants, QOBL-GA and QROBL-GA consistently show the best accuracy, particularly in the IP-SP configuration, while QROBL-ABC also excels in terms of STD across multiple configurations. In terms of execution time,

COOBL-GA and GOBL-PSO stand out as the fastest, particularly in the SP and IP-SP configurations, making them more efficient in terms of convergence speed. However, no variant demonstrates a clear improvement in NFEs, indicating that while some configurations achieve high accuracy, they do so at the expense of computational efficiency. Ultimately, while certain OBL-enhanced variants like QOBL-GA and QROBL-GA offer strong accuracy, the overall inefficiency in terms of function evaluations limits their practical application, particularly in large-scale, high-dimensional problems.

To conduct a rigorous comparative analysis of the five OBL variants applied to the five selected MAs at three integration phases, we employ Friedman's test—a robust non-parametric statistical method designed for evaluating multiple algorithms across diverse datasets. This test systematically ranks the algorithms based on their performance across multiple benchmark functions and determines whether the observed differences are statistically significant.

Tables S11–S15 present the Friedman mean rank results for the mean fitness values of DE, GA, PSO, HS, and ABC variants, respectively. The results indicate that the DE-QROBL-IP-SP configuration achieves the highest performance for $D = 10$, while DE-QROBL-SP emerges as the best-performing configuration for $D = 20$. Among the examined OBL techniques, QROBL and QOBL consistently attain the highest rankings for DE, GA, and HS at $D = 10$, and for DE, GA, and ABC at $D = 20$, demonstrating their superior effectiveness in enhancing algorithmic performance. In contrast, GOBL proves to be the most effective strategy for PSO, highlighting its adaptability and impact within this optimization framework.

### Convergence analysis

This section provides a comprehensive analysis and discussion of convergence curves for the five considered MAs and their OBL variants. The evaluation is conducted across four representative benchmark functions $F_1$, $F_3$, $F_7$, and $F_9$. Each function is selected from a different category—unimodal, multimodal, hybrid, and composition, in 10 and 20 dimensions. It is worth pointing out that we opted not to include all 12 functions to avoid making the article excessively lengthy. These convergence curves provide a visual representation of how the objective function evolves over iterations, allowing us to assess the algorithm's performance, solution quality, convergence speed, and balance between exploration and exploitation. We also detect issues like premature convergence and aid in comparative analysis. For a more detailed examination, all convergence diagrams are provided in the Supplemental File.

Figure S1, illustrates the convergence behaviour of DE and its variants during the initialization phase for $D = 10$. In $F_1$, all variants exhibit rapid convergence within the first 300 iterations, after which they stabilize at the same fitness value. Among them, DE and DE-QROBL demonstrate slightly faster convergence, reaching lower fitness values earlier than the other variants. This suggests that QROBL effectively enhances the early search phase by guiding the population toward promising regions more efficiently. For $F_3$, all DE variants demonstrate rapid convergence within the first 100 iterations. This rapid decline in fitness during the initial iterations indicates an efficient exploration of the search space.

Notably, the DE-QROBL slightly outperforms the others in the final stages. In $F_7$, the initial 400 iterations show significant fitness value reductions across all variants, accompanied by noticeable fluctuations. These fluctuations suggest that the algorithms are actively exploring the search space. DE-BOBL and standard DE stabilize earlier than the others, indicating that BOBL improves the balance between exploration and exploitation, helping the algorithm settle into optimal regions more efficiently. In $F_9$, the convergence curves reveal noticeable improvements across all variants occurring quickly, signifying an effective exploratory phase. However, they stabilize around 100 iterations, converging to similar fitness values.

Figure S2, presents the convergence results of the DE algorithm and its variants during the swarming phase for $D = 10$. For $F_1$, all variants exhibit rapid and comparable convergence, with most of the improvements occurring within the first few iterations. This swift convergence leads to an early stabilization of the fitness curves at very low values. The early flattening of the curves suggests that the search space for $F_1$ is relatively simple, allowing the algorithms to transition quickly from exploration to exploitation. In $F_3$, the fitness values drop sharply within the first 100 iterations, after which all variants stabilize at similar levels. The small differences in convergence speed suggest that OBL techniques do not drastically affect performance during the swarming phase. However, minor improvements in the final stages indicate that QROBL and COOBL may contribute to a slightly better-refined solution. In $F_7$, the first 400 iterations show a significant decrease in fitness values, with noticeable fluctuations across the variants. Interestingly, while most variants stabilize, DE-COOBL appears to stagnate at a higher fitness value, suggesting that it struggles to refine solutions effectively during the swarming phase. On the other hand, DE-QOBL and DE-GOBL show a more consistent descent, indicating that these OBL strategies improve the late-stage search process. For $F_9$, the curves show similar and fast convergence by the first 50 stagnating early and settling into local optima due to insufficient exploration. However, DE-COOBL reaches the lowest fitness value, demonstrating its ability to enhance solution quality.

Figure S3, illustrates the convergence behaviour of DE and its variants during the IP-SP phase for $D = 10$. In $F_1$, all variants exhibit rapid convergence within the first 200 iterations. The fitness values drop sharply, demonstrating effective exploration. After this early phase, the curves stabilize and converge to optimal fitness values, indicating that the variants transitioned into exploitation. Notably, the convergence patterns of DE-QROBL, and DE-COOBL are faster than others, indicating that they can provide marginal improvements in convergence speed for unimodal functions. For $F_3$, all variants once again display fast convergence, particularly within the first 100 iterations. The curves show very similar behaviour, with the fitness values stabilizing early at almost the same point for all variants. The fast drop in fitness in the initial iterations suggests efficient exploration of the search space. For $F_7$, the curves show more diversity in convergence behaviour compared to $F_3$. Most variants explore the search space in the first 200 iterations, after which they transition into the exploitation phase, each stabilizing over a varying number of iterations. DE-COOBL shows fast convergence behaviour and stabilizes by 100 iterations indicating distinct exploration ability. DE-GOBL, on the other hand, requires more iterations to fully

exploit the search space, resulting in a slower convergence rate compared to the other variants. For $F_9$, the convergence behaviour is largely similar across all variants. They display rapid convergence within the first 50 iterations, settling early into local optima due to insufficient exploration, except for DE-COOBL, which converges to a lower value. The early stabilization suggests that all DE variants prematurely converge to optimal solutions without further improvements in fitness.

Figure S4, presents the convergence plots of the DE and its variants during the initialization phase for $D = 20$. In $F_1$, the convergence curves for all DE variants exhibit a rapid decline in fitness values during the initial iterations, signaling an effective early exploration phase. However, the early flattening of the curves is primarily due to the simplicity of the problem, which requires minimal exploration. In $F_3$, all DE variants show rapid convergence, achieving their fitness values within the first 200 iterations. After this point, the fitness values stabilize. For $F_7$, the convergence behaviour indicates a prolonged exploration phase within the first 400 iterations, followed by a transition into the exploitation phase. Most variants stabilize by around 600 iterations, this indicates that they improve exploration efficiency, helping to avoid premature convergence, with the standard DE again showing the best convergence results. In $F_9$, the algorithms exhibit a rapid decline in fitness within the first 100 iterations, indicating efficient exploration of the solution space. By the end of the optimization process, all variants converge to the same fitness value, suggesting no distinct advantage in convergence behavior. This outcome is attributed to the characteristics of the function, which guide all algorithms along a similar optimal trajectory with minimal deviation.

Figure S5, illustrates the convergence curves of DE and its variants during the swarming phase for $D = 20$. In $F_1$, all variants exhibit smoother, more flattened curves, indicating smaller incremental improvements and entering a prolonged exploitation phase. Notably, DE-COOBL distinguishes itself by converging to its best fitness value by iteration 200, showcasing faster convergence and a more efficient optimization process compared to the other variants. In $F_3$, all variants exhibit rapid convergence within the first 200 iterations, demonstrating an effective exploration of the search space. Following this initial phase, the algorithms stabilize at similar fitness values. Notably, DE-COOBL outperforms the other variants, closely followed by DE-QROBL, both of which deliver good convergence results. For $F_7$, the convergence curves exhibit a more gradual convergence behaviour, with stability being reached later in the optimization process. During the first 200 iterations, the curves change gradually, indicating an active exploration phase. Following this, the algorithms transition into the exploitation phase, with smaller, less frequent improvements. The figure clearly highlights that DE-COOBL outperforms the other variants, achieving the best convergence results. For $F_9$ all variants exhibit similar convergence behavior, achieving rapid progress within the first 50 iterations before stagnating, indicating premature convergence.

Figure S6, illustrates the convergence curves of the DE algorithm and its variants during the IP-SP phase for $D = 20$. In $F_1$, all the curves converge rapidly and exhibit similar performance, indicating a robust exploration phase during the initial iterations. Following this, the algorithms transition into a more thorough exploitation of the search space,

leading to subtle improvements in fitness. For $F_3$, the curves reflect an effective exploration of the search space within the first 200 iterations, after which they stabilize, with all variants closely aligned. However, DE-COOBL demonstrates a slight advantage, outperforming the other variants for this function. In $F_7$, the curves take longer to converge compared to $F_3$ starting with a rapid descent in the first 200 iterations followed by slow improvements, indicating a balanced transition between exploration and exploitation. Notably, DE-COOBL distinguishes itself by achieving earlier convergence, outperforming the other variants. For $F_9$, the convergence curves exhibit a rapid trend, stabilizing within the first 50 iterations. DE-QROBL achieves a slightly faster convergence rate, reaching a lower fitness value ahead of the other variants.

Figure S7, illustrates the convergence curves of the GA algorithm and its variants during the Initialization phase for $D = 10$. In $F_1$, DE-BOBL starts with a relatively high fitness value but quickly catches up to the other variants. The leveling off of the curves during the initial iterations indicates a swift transition into the exploitation phase. In this phase, the algorithms exhibit only minor improvements in fitness before stabilizing later on. For $F_3$, all curves exhibit rapid convergence behaviour, stabilizing within the first 50 iterations in the same fitness value. However, the standard GA and GA-COOBL variants require additional iterations before achieving stability, indicating a more gradual refinement in their convergence process. Similarly, in $F_7$, the curves show a rapid convergence pattern, with fitness values decreasing significantly within the initial iterations. GA-QROBL has a fast convergence speed compared to the other variants. In $F_9$, we distinguish the standard and GA-QOBL the standard GA and GA-QOBL variants require more iterations to converge to their optimal fitness values and achieve stabilization. This extended convergence time suggests that these variants engage in a more thorough exploration phase, allowing them to search the solution space more comprehensively before settling on the final solution.

Figure S8, illustrates the convergence behaviour of the GA algorithm and its variants during the swarming phase for $D = 10$. In $F_1$, all the variants display uneven convergence speeds, marked by fluctuating fitness values during the initial iterations. Within the first 200 iterations, the algorithms primarily focus on exploring the search space before transitioning to the exploitation phase. Notably, GA-QROBL exhibits the fastest convergence, stabilizing at an early stage and effectively optimizing performance compared to the other variants. For $F_3$, all curves exhibit similar convergence patterns, showing rapid improvements within the first 30 iterations. After this, they stabilize at nearly the same fitness values, except for the standard GA, which takes about 200 iterations to stabilize. The early stabilization across most variants indicates premature convergence, as little to no improvement is seen after the initial 200 iterations. In $F_7$, the convergence behaviour is more gradual, with a slower decline in fitness values. Stabilization occurs after about 250 iterations. The slower transition into exploitation allows for more thorough exploration before convergence. GA-BOBL, followed by GA-QOBL, achieves the best results, demonstrating superior performance compared to the other variants. Finally, for $F_9$, the curves show rapid convergence within the first 50 iterations, with the exception of standard GA and GA-QOBL, which take more time to stabilize. After the initial convergence, the

curves flatten. The early convergence and the quick stabilization hints at possible premature convergence, as the algorithms may not have fully explored the search space. Expect GA-COOBL that outperforms the other variants, achieving the lowest fitness value.

Figure S9, presents the convergence plots of the GA and its variants during the combined IP-SP phase for $D = 10$. For $F_1$, all GA variants exhibit rapid convergence, with most achieving near-optimal solutions at iteration 200. This demonstrates the high efficiency of the GA variants in optimizing the unimodal function, with only slight performance differences observed among them. In $F_3$ the convergence occurs rapidly within the first 50 iterations, with all GA variants' performance appearing better than the standard GA. This indicates an enhanced ability of the variants to optimize this function effectively. For $F_7$, the convergence process is rapid for most variants, except for DE-QOBL, which requires more iterations to effectively exploit the search space. Most GA variants stabilize in 200 iterations and converge to similar fitness values, indicating a shared capacity for refinement within the solution space. In $F_9$, the algorithms demonstrate a rapid decline in fitness within the first few iterations indicating effective exploration of the solution space. Among the variants, DE-COOBL demonstrates superior performance, converging slightly faster than the others and achieving lower fitness values by the conclusion of the optimization process.

Figure S10, presents the convergence curves of the GA algorithm and its variants through the initialization phase for $D = 20$ space. In $F_1$ all variants demonstrate rapid and comparable convergence behaviour, with GA-QROBL and GA-QOBL stabilizing within the initial iterations. The other variants, however, show minor differences, indicating a brief exploitation phase that extends up to around 200 iterations. For $F_3$, the curves exhibit a gradual and distinct convergence, reaching stability around 400 iterations. During the first 200 iterations, the curves reflect an effective exploration phase, followed by a transition into the exploitation phase, where only slight improvements are observed. Notably, GA-COOBL and GA-GOBL outperform the other variants for this function, demonstrating superior convergence behaviour. In $F_7$, the rapid initial descent in the curves signifies a robust exploration phase, during which the algorithms quickly enhance the fitness values. However, the subsequent flattening of the curves after 200 iterations marks the transition into the exploitation phase, with slower, incremental improvements. GA-QROBL, in particular, stands out by demonstrating a strong convergence performance, reaching optimal fitness more effectively than the other variants. In the $F_9$ function, all variants demonstrate rapid convergence, reaching similar fitness values within the first 50 iterations, with no significant improvements afterward. This behaviour reflects a swift exploration phase followed by poor exploitation.

Figure S11, illustrates the convergence behaviour of the GA algorithm and its variants during the Swarming phase for $D = 20$. In $F_1$, all variants display rapid and closely aligned convergence rates, with their curves flattening early. This rapid stabilization indicates that the algorithms reach their optimal fitness values quickly. In contrast, in $F_3$, the curves demonstrate effective exploration during the initial 200 iterations, after which the algorithms Switch into the exploitation phase, gradually converging towards stabilization. GA-COOBL and GA-BOBL show slightly superior convergence rates compared to the

other variants, indicating better optimization performance throughout the process. Similarly for $F_7$, throughout the first 200 iterations, the curves reflect efficient exploration followed by a shift into the exploitation phase, where the algorithms progressively converge toward stability. Notably, GA-GOBL stands out by outperforming the other variants, demonstrating positive convergence behaviour. For $F_9$, the convergence behaviour of all variants exhibited a similar and rapid pace, with stabilization occurring within the initial 50 iterations. This rapid stabilization indicates that the algorithms may have converged prematurely.

Figure S12, illustrates the convergence curves of the GA algorithm and its variants during the IP-SP phase for $D = 20$. In $F_1$, GA-BOBL begins with a relatively high fitness value, rapidly decreasing to align with the other variants, displaying swift convergence during the initial iterations. All variants quickly attain the same fitness value without any subsequent improvements. For $F_3$, the difference between the variants is somewhat evident. All variants initially focus on exploring the search space for the first 200 iterations, after which they transition into the exploitation phase. Notably, GA-BOBL and GA-QOBL achieve the fastest convergence during this process, demonstrating their effectiveness in this function. In $F_7$, all the variants exhibit strong convergence behaviour, particularly during the initial 200 iterations, after which they stabilize around a similar fitness value. Furthermore, GA-COOBL demonstrates superior convergence performance, closely followed by GA-QOBL, which also yields strong results and attains the near-optimal fitness value. In $F_9$, all variants exhibit a steep convergence trend, rapidly approaching the same fitness value within the first 50 iterations. However, the lack of further improvement beyond this point indicates poor exploration and a strong tendency toward premature convergence.

Figure S13, illustrates the convergence curves of the PSO algorithm and its variants during the initialization phase for $D = 10$. In $F_1$, all variants demonstrate comparable convergence behaviour, displaying rapid progress during the first 100 iterations before approaching the same lower fitness value. In $F_3$, the variants again exhibit similar convergence performance, stabilizing at around 100 iterations, indicating an effective exploration phase. Notably, PSO-QROBL displays a slight advantage over the other variants. In $F_7$, the curves exhibit more distinct patterns, characterized by rapid convergence during the initial 100 iterations. This convergence is marked by frequent fluctuations in fitness values, highlighting an effective exploration phase. Following this period, the curves stabilize after 100 iterations to the optimal value. Notably, PSO-QOBL demonstrates particularly swift convergence, outperforming the other variants in this function. In $F_9$, the variants exhibit rapid convergence, making significant progress within the first 100 iterations. However, they stabilize too quickly without further refinement, indicating premature convergence.

Figure S14, illustrates the convergence curves of the PSO algorithm and its variants during the swarming phase $D = 10$. For $F_1$, PSO-COOBL starts at a comparatively high fitness value but swiftly declines to match the performance of the other variants, which also demonstrates quick convergence in the early iterations. All variants soon reach the same lower fitness level. FOR $F_3$, the curves exhibit strong convergence during the first 100

iterations, after which they stabilize at the optimal fitness value. However, PSO-COOBL gets stuck in a local optimum. Among the variants, PSO-QROBL achieves the fastest convergence rate, outperforming the other algorithms on this function. In $F_7$, the variants display a noticeable distinction in their convergence curves. All curves settle within the first 100 iterations in a local optimum, indicating premature convergence and insufficient exploration of the search space, which prevents further improvement toward the global optimum. In $F_9$, demonstrate rapid initial progress within the first 50 iterations, followed by stagnation after 100 iterations at a fitness value around 2,400–2,500, failing to reach the global optimum. This behavior indicates premature convergence, where the algorithms get trapped in a suboptimal solution due to insufficient exploration.

Figure S15, illustrates the convergence curves of the PSO algorithm and its variants during the IP-SP phase for $D = 10$. In $F_1$, the curves demonstrate rapid convergence in the first 100 iterations, and they all settle in the same fitness value. Notably, PSO-QROBL slightly outperforms others. For $F_3$, the curves once again exhibit rapid convergence behaviour within the first 100 iterations to the optimal solution, indicating strong exploration capabilities. PSO-COOBL stabilizes slightly earlier than the other variants, reflecting its effective optimization performance during this phase. Similarly, for $F_7$, all variants converge to the same fitness value within the first 100 iterations, indicating limited exploration of the search space. However, PSO-COOBL exhibits a faster convergence rate but ultimately stagnates in a local optimum, failing to achieve further improvements. In $F_9$, all curves exhibit comparable convergence behaviour, characterized by rapid convergence within the first 100 iterations. Following this initial phase, the curves stabilize at the same fitness value, indicating that the variants reach a similar level of optimization.

Figure S16, illustrates the convergence curves of the PSO algorithm and its variants during the initialization phase for $D = 20$. In $F_1$, PSO-BOBL starts with a relatively high fitness value but quickly aligns with the other PSO variants. All variants demonstrate swift convergence during the initial iterations, showcasing strong early exploration capabilities. Furthermore, the curves for all variants flatten out and stabilize at the same fitness value. For $F_3$, the variants exhibit swift convergence within the initial 100 iterations, with PSO-GOBL showing a slight advantage by stabilizing marginally earlier than the others. This rapid initial convergence reflects strong exploration capabilities, while the early stabilization suggests an expedited shift into the exploitation phase. In $F_7$ the convergence curves exhibit noticeable distinctions within the first 200 iterations, with all variants demonstrating fast progress. However, despite the rapid improvement, they ultimately settle at a lower yet suboptimal fitness value, highlighting challenges in effective exploitation. For $F_9$, all variants exhibit a swift and nearly complete convergence within the initial 100 iterations, quickly stabilizing at a suboptimal fitness value, Indicating a premature shift from exploration to exploitation.

Figure S17, illustrates the convergence behaviour of the PSO algorithm and its variants during the swarming phase for $D = 20$. In $F_1$, PSO-BOBL once again starts with a higher initial fitness value but quickly matches the performance of the other PSO variants. All variants exhibit rapid convergence in the early iterations, reflecting strong exploration capabilities. For $F_3$, all variants converge within the initial 100 iterations, exhibiting

comparable convergence behaviour as they settle at a similar fitness value. However, PSO-COOBL stands out by converging to a slightly higher fitness value. During the first 100 iterations, all variants demonstrate clear exploration of the search space, showcasing their ability to identify promising solutions before stabilizing. In $F_7$, the variants exhibit distinct convergence curves, beginning with an exploration phase of the search space before transitioning to exploitation after 100 iterations. Among them, PSO-COOBL demonstrates a fast convergence performance, closely trailed by the PSO-QROBL variant. In $F_9$, all curves show similar convergence behaviour. They show a fast convergence by the first 100 iterations then they stabilize to the same suboptimal fitness value.

Figure S18, depicts the convergence behaviour of the PSO algorithm and its variants during the IP-SP phase for $D = 20$. In $F_1$, PSO-COOBL begins with a higher fitness value, however, it rapidly converges to match the performance level of the other variants within just a few iterations. All PSO variants converge extremely fast within the first few iterations and stabilize, with little to no further changes as iterations progress. This quick stabilization of the curves comes from the simple complexity of the unimodal function that requires minimal exploration. For $F_3$, all variants converge within the first 100 iterations. The small variations in the early iterations between the PSO variants, with PSO-QROBL showing the most effective convergence to the lowest fitness value earlier. In contrast PSO-COOBL stagnates in a local optimum, failing to achieve further improvement. For $F_7$, the convergence curves exhibit more variation compared to those in $F_3$. PSO-COOBL begins with a significantly higher fitness value but quickly aligns with the performance of the other variants indicating a distinctive exploration strategy in navigating the search space. Despite the differing starting points, all variants ultimately converge toward a similar fitness level. PSO-QROBL attains a fast yet lower fitness value. For $F_9$, all variants exhibit rapid convergence, typically within the initial 100 iterations. The convergence curves show minimal variation, with most settling around a similar fitness value. This flattening of the curves suggests that the algorithms shift to exploitation too quickly, thereby restricting their ability to achieve further improvements.

Figure S19, illustrates the convergence curves of the HS algorithm and its variants during the initialization phase for $D = 10$. For $F_1$, it is marked by rapid decreases in fitness values during the initial iterations for all variants, with most of the improvement happening within the first 200 iterations. HS-QOBL and HS-QROBL exhibit the fastest convergence, with the steepest decline in fitness values. Both reach the lowest fitness values relatively early and show very little change after approximately 200 iterations. This indicates a strong early exploration capability, followed by efficient exploitation. For $F_3$, all variants show a rapid decrease in fitness values during the first 200 iterations, signalling strong exploration during the early phases. Afterwards, the curves level off, indicating the transition into exploitation. HS-GOBL achieves a slightly better convergence performance. For $F_7$, all variants experience rapid decreases in fitness values early in the iterations, similar to the previous functions, except HS-BOBL which needs an extended exploration. Once again, HS-QROBL and HS-QOBL lead in convergence, reaching the best fitness value early on and showing minimal change after around 200 iterations. In $F_9$, all curves exhibit a steep decline in fitness values within the first 200 iterations, after which they

plateau as the algorithms shift into the exploitation phase. However, this stabilization occurs before fully reaching the most promising regions, indicating potential limitations in exploration.

Figure S20, illustrates the convergence curves of the HS algorithm and its variants during the swarming phase for $D = 10$. In $F_1$, all HS variants exhibit rapid convergence during the initial 200 iterations, with HS-QROBL and HS-QOBL demonstrating the most pronounced decline in fitness values. After this point, the convergence rate slows significantly, leading to stabilization in the best fitness values, which suggests efficient exploration during the early phase. For $F_3$, all HS variants show comparable convergence behaviour, achieving rapid convergence within the first 200 iterations. Each variant performs effectively, reaching nearly best fitness values, with minimal variation among them. However, HS-QOBL exhibits a slightly faster convergence rate compared to the others. In $F_7$, the convergence curves display more noticeable variation. HS-COOBL initially starts with a higher fitness value but eventually surpasses the others in terms of convergence within the first 100 iterations. In contrast, HS-GOBL demonstrates a slower convergence rate, indicating a prolonged exploitation phase. For $F_9$, all variants exhibit a uniform convergence pattern, rapidly reducing fitness values within the first 100 iterations before stabilizing at a slightly higher fitness value, indicating insufficient exploration. Notably, HS-COOBL follows a similar convergence trend but achieves a lower final fitness value.

Figure S21, illustrates the convergence curves of the HS algorithm and its variants during the IP-SP phase for $D = 10$. For $F_1$, HS-BOBL initially starts with a significantly higher fitness value compared to the other variants but quickly catches up, demonstrating strong early exploration. All HS variants show rapid convergence within the first 200 iterations, with HS-QROBL displaying the fastest convergence. After 200 iterations, the rate of convergence slows considerably, and the fitness values stabilize, indicating a shift to the exploitation phase. For $F_3$, all variants are equally effective at reaching low and similar final fitness values. All of them converge quickly, within the first 200 iterations. The exploration is very efficient in the early iterations, as evidenced by the sharp decline. However, after that, the algorithms enter the exploitation phase and show weak improvement. In $F_7$, there is a more pronounced variation in the performance of the HS variants. HS-QROBL again shows the fastest convergence, quickly attaining the best fitness value. The exploration phase is strong for HS-QROBL during the first 200 iterations, after which the convergence curves flatten, marking the transition to exploitation. For $F_9$, all variants exhibit a convergence pattern similar to other phases, rapidly approaching a suboptimal fitness value within the first 100 iterations and showing minimal improvement thereafter. Notably, HS-COOBL achieves the lowest final fitness value, highlighting its Better exploitation capability for this function.

Figure S22, illustrates the convergence curves of the HS algorithm and its variants during the initialization phase for $D = 20$. In $F_1$, HS-BOBL begins with a significantly higher fitness value but rapidly reaches the other variants. All HS variants converge to optimal fitness values within the early iterations, after which their convergence curves stabilize and flatten. For $F_3$, all variants demonstrate comparable effectiveness, reaching

similarly the best fitness values. They converge swiftly, completing most of the process within the first 200 iterations. The early iterations exhibit highly efficient exploration, reflected by the steep drop in fitness values. However, in the exploitation phase, the curves show only slight improvements. For $F_7$, all the variants exhibit fast convergence within the first 200 iterations, indicating a robust exploration phase. Afterwards, the curves show slight improvements, marking the transition into the exploitation phase. This gradual improvement continues until all variants stabilize in the near-optimal fitness value, with convergence completed by approximately 600 iterations. For $F_9$, all variants exhibit rapid convergence within the first 50 iterations, quickly stabilizing at a consistent yet suboptimal fitness value. The steep early decline suggests insufficient exploration which hinders further improvement beyond this point.

Figure S23, shows the convergence curves of HS and its variants during the swarming phase for $D = 20$. In $F_1$, all variants demonstrate extremely rapid convergence, with the fitness values stabilizing within the first 50 iterations. There is no notable difference in either convergence speed or final fitness values among the variants, as the fitness value drops quickly to near-zero levels. This suggests that the problem is relatively simple, leaving no significant need for extended exploration. In $F_3$, all curves take approximately 250 iterations to reach their best fitness value, indicating a well-balanced combination of exploration and exploitation capabilities. In $F_7$, there is a clearer distinction between the performance of the different variants. The convergence curves reduce higher fitness values in 200 iterations, showcasing an effective exploration phase. Following this, the algorithms transition into exploitation, progressively stabilizing. HS-COOBL stands out by converging to a lower fitness value compared to the other variants, demonstrating superior performance. For $F_9$, all variants exhibit rapid convergence within the initial 100 iterations, after which they stabilize at a fitness value slightly above 2,500. This behavior suggests efficient initial exploration but limited exploitation, as none of the variants manage to reach the optimal fitness value of 2,300.

Figure S24, illustrates the convergence behaviour of HS and its variants during the IP-SP phase for $D = 20$. In $F_1$, all variants converge very quickly within the first 20 iterations. The fitness values start high but drop sharply to near-zero values. This sharp drop indicates an efficient initial exploration phase. However, the swift transition into exploitation suggests that the algorithms required minimal extended exploration, leading to their rapid convergence and stabilization at optimal or near-optimal fitness values. In $F_3$, all algorithms display comparable convergence rates, with no single variant showing a clear advantage in either convergence speed or the final fitness value achieved. The curves show a good convergence process. Stabilization occurs around 300 iterations demonstrating an effective exploration. In $F_7$, the convergence curves exhibit notable distinctions, with a clearly defined exploration phase lasting through the first 200 iterations. The algorithms maintain higher fitness values during this phase, effectively searching the solution space. After this period, they transition into the exploitation phase, where the curves stabilize around 400 iterations in near-optimal value. Among the variants, HS-COOBL stands out for its faster convergence, reaching a lower fitness value more efficiently than the others. In $F_9$, all variants exhibit similar convergence patterns, with a rapid approach toward optimal

fitness within the first 50 iterations. Their swift convergence highlights effective early exploration, but limited exploitation, as none of the variants manage to reach the optimal fitness value.

Figure S25, illustrates the convergence curves of the ABC algorithm and its variants during the initialization phase for $D = 10$. In $F_1$, distinct differences in the convergence patterns of the ABC variants are observed. During the first 200 iterations, most of the variants demonstrate significant exploration, as indicated by the rapid drops in fitness values, before transitioning into the exploitation phase, where improvements become more incremental. However, both the standard ABC and ABC-COOBL fail to converge, as their fitness values continue to fluctuate even after 1,000 iterations. Among the variants, ABC-QROBL exhibits the best performance, achieving lower fitness values more efficiently, suggesting superior convergence. For $F_3$, all variants exhibit nearly identical convergence behaviour. The curves show a rapid decline within the first 100 iterations, indicating a strong exploration and exploitation capability across all algorithms. In $F_7$, all the variants demonstrate significant variation in their convergence patterns. The exploration phase occurs within the first 300 iterations, where the curves converge rapidly, followed by a slower progression as they enter the exploitation phase. ABC-QROBL shows faster convergence, refining the solution more effectively. In $F_9$, the curves start with a high initial fitness value. all variants show rapid convergence within the first 200 iterations. There is minimal variation in the exploration phase across the algorithms, with ABC-QROBL still slightly outperforming the others. However, the standard ABC algorithm shows weaker performance during the initial phase and continues to improve gradually throughout the iterations.

Figure S26, illustrates the convergence results of the ABC algorithm and its variants during the swarming phase for $D = 10$. In $F_1$, most variants exhibit notable variations in convergence behaviour, with significant fluctuations characterized by sharp rises and falls in fitness values. This instability suggests that the algorithms frequently shift between exploration and exploitation, potentially struggling to maintain consistent solution refinement within the search space. However, ABC-COOBL and ABC-QROBL stand out by demonstrating a much smoother convergence pattern, steadily improving their solutions and ultimately achieving the best performance among the variants. In $F_3$, the convergence curves for all variants display rapid improvements within the first 100 iterations, indicating that the algorithms quickly identify promising regions of the search space and transition efficiently toward optimal solutions. In $F_7$, all variants exhibit significant fluctuations during the first 300 iterations, which then become minimal in the later stages, and all variants converge to relatively slightly high fitness values. ABC-COOL once again outperforms the others, demonstrating superior convergence and achieving better performance in the optimization process. In $F_9$, the variants display distinct convergence behaviour during the first 200 iterations, reflecting a strong exploration effort. After this phase, they transition into exploitation. ABC-COOBL excels in terms of convergence speed.

Figure S27, illustrates the convergence patterns of the ABC algorithm and its variants during the IP-SP phase for $D = 10$. In $F_1$, the standard ABC, ABC-BOBL, and ABC-GOBL

show notable fluctuations, marked by sharp increases and decreases in fitness values. This instability implies that the algorithms often alternate between exploration and exploitation phases, possibly struggling to consistently refine solutions within the search space. In contrast, ABC-COOBL, ABC-QROBL, and ABC-QOBL demonstrate a much smoother convergence process, gradually improving their solutions and ultimately delivering the best performance in terms of both convergence speed and fitness value. In $F_3$, all variants demonstrate rapid convergence within the first 100 iterations, indicating that the algorithms quickly pinpoint promising regions of the search space and efficiently shift toward optimal solutions. This swift convergence reflects strong early exploration followed by effective exploitation. For $F_7$, all variants display some fluctuations throughout the optimization process, indicating instability between the exploration and exploitation phases. Notably, most variants converge to a higher fitness value. However, ABC-COOL stands out, demonstrating superior convergence and stability. It effectively navigates the search space, consistently achieving better performance and producing more optimal results compared to the other variants. For $F_9$, all the variants exhibit distinct convergence patterns during the initial 200 iterations, indicating a robust exploration phase. After this stage, they shift into the exploitation phase. ABC-COOBL stands out by excelling in both convergence speed and fitness value, reinforcing its superiority over the other variants.

Figure S28, presents the convergence curves of ABC and its variant during the initialization phase for $D = 20$. In $F_1$, most variants exhibit predominantly flat convergence curves over the majority of iterations, signalling a common tendency toward early stabilization. The standard ABC and ABC-COOBL, in particular, show notably poor convergence rates, reflecting inefficiencies in their optimization process. Both struggle to refine their solutions effectively, resulting in suboptimal performance and higher final fitness values. ABC-GROBL and ABC-QOBL, while also showing a flat curve, indicate stagnation early on, with little to no improvement in fitness values as the iterations progress, underscoring its limited exploration and exploitation balance. In $F_3$, all variants exhibit rapid convergence within the first 300 iterations. The fitness values drop sharply during this phase, which suggests a strong exploration ability by all algorithms in quickly identifying promising regions of the search space. After this initial phase, the curves for all variants become closely aligned, indicating that they enter the exploitation phase around the same time. In $F_7$, the convergence behaviour of the variants shows greater variation, with notable fluctuations during the first 200 iterations, signalling active exploration of the search space. The standard ABC initially distinguishes itself by outperforming all other variants in these early stages. However, as the optimization progresses, ABC-QOBL takes the lead, excelling in the latter stages by achieving the lowest fitness value, and demonstrating superior exploitation capabilities in refining solutions toward the optimum. In $F_9$, all variants effectively explore the search space and converge to the suboptimal during the first 200 iterations. After this initial phase, the curves level out, indicating a transition into the exploitation phase. Notably, ABC-QROBL demonstrates the best convergence, rapidly reducing fitness values ahead of the other variants.

Figure S29, illustrates the convergence curves of the ABC algorithm and its variants during the swarming phase for $D = 20$. In $F_1$, all variants experience a rapid decline in

fitness values, followed by predominantly flat convergence curves throughout most of the iterations, indicating a shared tendency toward early stabilization in their optimization processes. In $F_3$, a rapid decline in fitness values occurs during the first 300 iterations, signaling a strong exploration phase. All variants identify promising areas of the search space early on, which contributes to their steep initial convergence. After the first 300 iterations, the curves begin to flatten, indicating a transition into the exploitation phase. ABC-COOBL followed by ABC-QROBL demonstrate better performance in terms of faster convergence and reaching lower fitness values compared to the other variants. For $F_7$, a greater variation in the convergence behaviour of the variants is observed, with some irregular fluctuations, indicating that the algorithms switch between exploration and exploitation in the search space. ABC-COOBL stands out by consistently outperforming all other variants, particularly in achieving the lowest fitness value during the early stages of optimization. In $F_9$, all variants explore the search space during the first 200 iterations. After this initial phase, the curves level out, indicating a transition to the exploitation phase. In particular, ABC-QROBL demonstrates a slight best convergence, rapidly reducing fitness values ahead of the other variants.

Figure S30, presents the convergence behaviour of the ABC and its variants during IP-SP phase for $D = 20$. For $F_1$, ABC-BOBL, ABC-GOBL and ABC-QOBL exhibit some oscillatory behaviours, with their fitness values fluctuating throughout the iterations. This indicates an inconsistent performance, with the algorithms alternating between exploration and exploitation. ABC-QROBL and ABC-COOBL show much smoother convergence with a steady decline in fitness values. They exhibit the fastest and most stable convergence behaviour, indicating strong performance in both the exploration and exploitation phases. In $F_3$, all algorithms demonstrate a steep initial decline in fitness values during the first 300 iterations, indicating a strong exploration phase. After this, the convergence rate slows, reflecting the transition into the exploitation phase. ABC-COOBL outperforms the other variants by achieving the lowest fitness values earlier. In $F_7$, ABC-COOBL stands out by quickly reducing the fitness values in the early iterations and stabilizing at a lower fitness level, demonstrating its superior ability to find good solutions. In contrast, while other variants demonstrate relatively competitive performance during the initial iterations, later on, their curves exhibit more fluctuations, suggesting some instability and inefficiency in their search process. For $F_9$, all variants exhibit rapid initial convergence within the first 200 iterations, followed by a gradual, steady decline in fitness values. ABC-COOBL outperforms the others, achieving the lowest fitness value. However, all variants ultimately stabilize near the optimal fitness value.

## CONCLUSION AND PERSPECTIVES

In conclusion, this study explored the integration of five opposition-based learning—BOBL, QOBL, GOBL, COOBL, QROBL—with five prominent MAs: DE, GA, PSO, ABC, and HS. By applying these OBL variants to the CEC 2022 benchmark functions, the primary objective was to assess their impact on enhancing the performance of metaheuristics in solving complex optimization problems.

Our analysis, leveraging key performance indicators such as maximum, minimum, mean, standard deviation, and convergence curves, provided a comprehensive evaluation of algorithmic efficacy. Additionally, the Friedman test was employed to statistically substantiate performance disparities among the algorithmic variants. The findings reveal that QROBL consistently outperforms other OBL variants across the majority of benchmark functions, exhibiting superior convergence speed and enhanced solution quality. Furthermore, COOBL and QOBL demonstrated substantial performance enhancements over their respective baseline algorithms, reinforcing the critical role of OBL strategies in advancing metaheuristic optimization.

The results confirm that OBL techniques, particularly QROBL, can significantly enhance the capability of MAs in addressing complex optimization challenges. Future research can extend this work by investigating additional benchmark functions, refining OBL techniques, and exploring their applications in real-world optimization problems, where traditional methods often encounter limitations.

Despite these promising results, several limitations must be acknowledged, along with potential solutions. OBL variants typically require higher NFEs compared to their original counterparts, leading to increased computational costs. To mitigate this, adaptive function evaluations could be integrated to optimize computational efficiency. Additionally, the influence of the jumping rate in generation updates plays a crucial role in performance, necessitating careful fine-tuning. A possible solution is the incorporation of self-adaptive jumping rates to enhance robustness across different problem landscapes. Furthermore, the reliance on fixed parameter settings may hinder adaptability, emphasizing the need for adaptive parameter control mechanisms.

### Funding
This research is funded by Ongoing Research Funding program, (ORF-2025-809) (previously known as Researchers Supporting Program (RSPD2025R809)), King Saud University, Riyadh, Saudi Arabia. The funders had no role in study design, data collection and analysis, decision to publish, or preparation of the manuscript.

### Grant Disclosures
The following grant information was disclosed by the authors:
Ongoing Research Funding program: ORF-2025-809, previously known as Researchers Supporting Program (RSPD2025R809).
King Saud University, Riyadh, Saudi Arabia.

### Competing Interests
The authors declare that they have no competing interests.

## Author Contributions

- Rihab Lakbichi conceived and designed the experiments, performed the experiments, analyzed the data, performed the computation work, prepared figures and/or tables, authored or reviewed drafts of the article, software, and approved the final draft.
- Farouq Zitouni conceived and designed the experiments, performed the experiments, analyzed the data, performed the computation work, prepared figures and/or tables, authored or reviewed drafts of the article, software, Supervision, and approved the final draft.
- Saad Harous conceived and designed the experiments, performed the experiments, analyzed the data, prepared figures and/or tables, authored or reviewed drafts of the article, supervision, and approved the final draft.
- Aridj Ferhat conceived and designed the experiments, performed the experiments, analyzed the data, prepared figures and/or tables, authored or reviewed drafts of the article, software, and approved the final draft.
- Abdelhadi Limane conceived and designed the experiments, performed the experiments, analyzed the data, prepared figures and/or tables, authored or reviewed drafts of the article, software, and approved the final draft.
- Abdulaziz S. Almazyad analyzed the data, authored or reviewed drafts of the article, and approved the final draft.
- Ali Wagdy Mohamed analyzed the data, authored or reviewed drafts of the article, and approved the final draft.

## Data Availability

The raw data and code are available in the Supplemental File.

## Supplemental Information

Supplemental information for this article can be found online at http://dx.doi.org/10.7717/peerj-cs.2935#supplemental-information.

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
