# Peer review of "Opposition-based learning techniques in metaheuristics: classification, comparison, and convergence analysis"

_PeerJ Computer Science, doi:10.7717/peerj-cs.2935_

## Round 0.1 · original submission · Major Revisions

Dear authors,

Reviewers have now commented on your article. We do encourage you to address the concerns and criticisms of the reviewers with respect to reporting, experimental design, and validity of the findings and resubmit your article once you have updated it accordingly. Following should also be addressed:

1. All equations should be used with correct equation number. Equations should be used with correct equation number. Please do not use “as follows”, “given as”, etc. Explanation of the equations should be checked. Definitions and boundaries of all variables should be provided. Necessary references should also be given.
2. Many of the equations are part of the related sentences. Attention is needed for correct sentence formation.
3. Please pay special attention on the usage of abbreviations.
4. Reviewer 2 has asked you to provide specific references. You are welcome to add them if you think they are useful and relevant. However, you are under no obligation to include them, and if you do not, it will not affect my decision.

Best wishes,

Reviewer 1 ·

Basic reporting

This paper, the authors study comparison analyzes nine distinct variants opposition-based learning (OBL) techniques in metaheuristic optimization field's applications, the experiments were conducted using 12 CEC 2022.benchmark functions,the results show that the current optimum opposition-based learning consistently outperforms the other variants in enhancing the MAs' performance, demonstrating superior convergence speed and solution quality across most of the benchmark functions.The topic is indeed timely and of interest. Some of my concerns regarding this paper are as follows:
1. The literature review should focus on all the optimization algorithm for OBL’s the advantages and disadvantages in MAs. This would help the paper to have a better understanding of the real contributions of the paper.
2. What is the superiority of the authors' work in comparison with those of others work for OBL in metaheuristic optimization field's applications.
3. Authors should provide a table containing the features of each OBLs used to conclude on what is that that make the proposed approach superior.
4. Suggest that there should be a detailed analysis of the results, for example: why OBL in techniques in metaheuristic optimization field's applications can produce better results?
5. The OBL techniques in metaheuristic limitations should be discussed.
6. References lack a high level of journal references.
7. Suggest add discuss the challenges when applying to large-scale problems, especially by discussing how large problems can be tackled without significant problems (in terms of dimensions).
8.The convergence analysis section should be improved with deeper analysis.

Experimental design

The convergence analysis section should be improved with deeper analysis.

Validity of the findings

Suggest add discuss the challenges when applying to large-scale problems, especially by discussing how large problems can be tackled without significant problems (in terms of dimensions).

Additional comments

No.

Reviewer 2 ·

Basic reporting

1. The abstract part needs to explain the writing motivation and research process of the article more clearly. Authors should provide the ideas, strategies, and methods of the proposed method in detail.
2. In the introduction, the main contributions of your paper are not clear. Please further summarize and clearly demonstrate the main contributions of your paper as several points.
3. In Expriment, it should include a detailed comparison using metrics to provide a comprehensive performance evaluation of the proposed method.
4. In the Conclusion, could you tell me the limitations of the proposed method? How will you solve them? Please add this part to the manuscript.
5. The literature review is poor in this paper. I hope that the authors can add some new references in order to improve the reviews. For example, https://doi.org/10.1038/s41467-024-54069-5,https://doi.org/10.1016/j.cja.2022.08.020 and https://doi.org/10.1109/JSEN.2024.3516124 and so on.
6. Authors also need to work on Grammar and the use of the English Language.
7. Remove prolonged sentences, write simple sentences that are easy to read.

Experimental design

1. The abstract part needs to explain the writing motivation and research process of the article more clearly. Authors should provide the ideas, strategies, and methods of the proposed method in detail.
2. In the introduction, the main contributions of your paper are not clear. Please further summarize and clearly demonstrate the main contributions of your paper as several points.
3. In Expriment, it should include a detailed comparison using metrics to provide a comprehensive performance evaluation of the proposed method.
4. In the Conclusion, could you tell me the limitations of the proposed method? How will you solve them? Please add this part to the manuscript.
5. The literature review is poor in this paper. I hope that the authors can add some new references in order to improve the reviews. For example, https://doi.org/10.1038/s41467-024-54069-5,https://doi.org/10.1016/j.cja.2022.08.020 and https://doi.org/10.1109/JSEN.2024.3516124 and so on.
6. Authors also need to work on Grammar and the use of the English Language.
7. Remove prolonged sentences, write simple sentences that are easy to read.

Validity of the findings

1. The abstract part needs to explain the writing motivation and research process of the article more clearly. Authors should provide the ideas, strategies, and methods of the proposed method in detail.
2. In the introduction, the main contributions of your paper are not clear. Please further summarize and clearly demonstrate the main contributions of your paper as several points.
3. In Expriment, it should include a detailed comparison using metrics to provide a comprehensive performance evaluation of the proposed method.
4. In the Conclusion, could you tell me the limitations of the proposed method? How will you solve them? Please add this part to the manuscript.
5. The literature review is poor in this paper. I hope that the authors can add some new references in order to improve the reviews. For example, https://doi.org/10.1038/s41467-024-54069-5,https://doi.org/10.1016/j.cja.2022.08.020 and https://doi.org/10.1109/JSEN.2024.3516124 and so on.
6. Authors also need to work on Grammar and the use of the English Language.
7. Remove prolonged sentences, write simple sentences that are easy to read.

Reviewer 3 ·

Basic reporting

The writing in the manuscript still needs to be standardized, such as keyword selection, formula numbering, and so on.

Experimental design

The experimental comparison data is not detailed enough, and it is necessary to provide specific performance data of each algorithm under average conditions, and then provide comparative summary results.

Validity of the findings

no comment

Additional comments

1.The keywords should be 4-6, there is no need to list every type of reverse learning.
2.Lack of numerical annotation for formulas on page six.
3.This is a great proposal to include in the introduction section the number of articles published in the past 5 years in SCIE on the selected 5 well established MAs to reflect the research popularity of such algorithms.
4.Table 12 only conducted statistics and did not provide actual data, which is different from the maximum mentioned in the abstract, There are deviations in minimum, mean, standard deviation, and convergence curves.
5.The convergence diagram of some functions should be provided in this manuscript.
6.Time complexity analysis can be conducted on multiple OBLs to determine if there are any time differences.
7.The article only tested the 12 functions of CEC2022 and did not use them in the more commonly used CEC2017 function. In addition, the data seems to have not been subjected to Friedman's test. Is the conclusion drawn too directly?
8.The first reference is incorrect. Please check each reference to meet the requirements.

---

## Round 0.2 · accepted · Accept

Dear Authors,

Thank you for addressing the reviewer's concerns and criticisms. Your manuscript now seems sufficiently improved and ready for publication.

Best wishes,

Reviewer 1 ·

Basic reporting

This review on opposition-based learning techniques in metaheuristics has significant reference value.

Experimental design

To experimentally demonstrate the capability of OBL strategies to enhance MAs is effective.

Validity of the findings

Meet the journal standards.

Additional comments

I feel that the authors have responded to all the comments raised by reviewers, and the manuscript is appropriate for publication.

Reviewer 2 ·

Basic reporting

This paper can be accepted now.

Experimental design

This paper can be accepted now.

Validity of the findings

This paper can be accepted now.

Additional comments

This paper can be accepted now.